# Highly sensitive in vivo detection of dynamic changes in enkephalins following acute stress in mice

Marwa O Mikati[1,2,3], Petra Erdmann-Gilmore[4], Rose Connors[4], Sineadh M Conway[1,2,3], Jim Malone[4], Justin Woods[1,2,3], Robert W Sprung[4], Reid R Townsend[4], Ream Al-Hasani[1,2,3]*

[1]Department of Anesthesiology, Washington University in St. Louis, St Louis, United States; [2]Washington University Pain Center, Washington University in St. Louis, St. Louis, United States; [3]Center for Clinical Pharmacology, University of Health Science and Pharmacy, St. Louis, United States; [4]Department of Medicine, Washington University School of Medicine, Washington University in St. Louis, St Louis, United States

*For correspondence:
al-hasanir@wustl.edu

Competing interest: The authors declare that no competing interests exist.

## eLife Assessment

The authors adapt a previously-established method that permits detection of in vivo extracellular levels of two distinct enkephalin opioid peptides in response to stressful experiences in mice. The present study highlights the potential of measuring actual peptides by microdialysis-LC-MS. They use this approach in conjunction with fiber photometric calcium imaging to correlate enkephalin neuron activity and enkephalin release in response to repeated stress, providing **convincing** evidence that this improved approach can provide new insights into opioid signaling in-vivo. This **important** study provides a means to understand various behavioral states controlled by endogenous opioids and the nucleus accumbens, including hedonic and stress responses, in health and disease. This work will be of broad interest to the neuroscientific community.

**Abstract** Enkephalins are opioid peptides that modulate analgesia, reward, and stress. In vivo detection of enkephalins remains difficult due to transient and low endogenous concentrations and inherent sequence similarity. To begin to address this, we previously developed a system combining in vivo optogenetics with microdialysis and a highly sensitive mass spectrometry-based assay to measure opioid peptide release in freely moving rodents (Al-Hasani et al., 2018, eLife). Here, we show improved detection resolution and stabilization of enkephalin detection, which allowed us to investigate enkephalin release during acute stress. We present an analytical method for real-time, simultaneous detection of Met- and Leu-enkephalin (Met-Enk and Leu-Enk) in the mouse nucleus accumbens shell (NAcSh) after acute stress. We confirm that acute stress activates enkephalinergic neurons in the NAcSh using fiber photometry and that this leads to the release of Met- and Leu-Enk. We also demonstrate the dynamics of Met- and Leu-Enk release as well as how they correlate to one another in the ventral NAc shell, which was previously difficult due to the use of approaches that relied on mRNA transcript levels rather than posttranslational products. This approach increases spatiotemporal resolution, optimizes the detection of Met-Enk through methionine oxidation, and provides novel insight into the relationship between Met- and Leu-Enk following stress.

## Introduction

Endogenous opioid peptides are released at significantly lower concentrations than small molecule neurotransmitters and undergo posttranslational processing that yields ~30 unique peptides from 4 precursor molecules. Together, these properties have made sensitive peptide detection difficult to achieve with existing technologies (*Fricker et al., 2020*). Here, we describe an analytical method that enables quantification of dynamic in vivo changes in both Leu- and Met-enkephalin (Leu-Enk and Met-Enk) peptides in rodents. We show unique release profiles between Leu- and Met-Enk in response to two different stressors and gain new insight into peptide release dynamics; additionally, our method offers improved spatiotemporal resolution. More broadly, this approach will enable an understanding of how neuropeptide levels change in response to acute or chronic behavioral, pharmacological, and neurophysiological manipulations.

Enkephalins can act on both the delta and/or mu opioid receptors, but their activity often depends on receptor and peptide expression patterns, which differ across the brain (*Banghart et al., 2018*; *Gomes et al., 2020*; *Mansour et al., 1995*; *Banghart et al., 2015*). Confirming an opioid peptide's identity based on the activity of the receptor alone poses difficulties. For example, in a single brain region, the mu opioid receptor may be activated by an enkephalin or endorphin peptide (*Gomes et al., 2020*). Therefore, any technique that relies on a readout of mu opioid receptor activity is not sufficient to determine whether the peptide is an endorphin or an enkephalin. It has been particularly challenging to measure Met- and Leu-Enk selectively and independently in vivo; in fact, both peptides are typically referred to as 'enkephalins', which oversimplifies their distinct properties. Though antibody-based techniques (*Maidment et al., 1989*; *Marinelli et al., 2005*; *Hernandez and Hoebel, 1988*; *Nieto et al., 2002*) or liquid chromatography/mass spectrometry (LC-MS) *Mabrouk et al., 2011*; *Li et al., 2009*; *Baseski et al., 2005* have improved Met- and Leu-Enk detection somewhat, limitations such as spatiotemporal restrictions and poor selectivity between enkephalins remain. Our group previously developed real-time in vivo detection of evoked opioid peptide release using microdialysis/nanoLC-MS (nLC-MS) after photostimulation (*Al-Hasani et al., 2018*). From this strong foundation, we have now optimized a novel opioid peptide detection technique. For the first time, we can selectively measure real-time in vivo levels of basal (non-evoked) peptide release, as well as during behavioral manipulations. Our primary goals in establishing this analytical workflow to measure opioid peptide dynamics in vivo were to (1) increase temporal resolution allowing for the detection of dynamic changes to short external stimuli and acute conditions, (2) lower the detection threshold to allow for accurate measurements from small brain regions using less volume of sample, (3) measure endogenous release of peptides in the native state without the need for photo- or chemogenetic stimulation, (4) report raw values of measured concentrations rather than % baseline measurements, (5) enable real-time, simultaneous detection of both Met- and Leu-Enk. To meet these goals and measure both Leu- and Met-Enk, we used an improved and specialized microdialysis technique coupled with nLC-MS.

## Results

### Optimizing a method for the simultaneous in vivo detection of Met- and Leu-Enk

Our workflow (*Figure 1A*) starts with stereotaxic surgery to implant the microdialysis probe in the mouse nucleus accumbens shell (NAcSh). Along with others, we have shown that the opioid receptor system in the NAcSh mediates both reward and aversion through photostimulation of peptide-expressing neurons across the dorsal/ventral axis (*Al-Hasani et al., 2015*; *Pirino et al., 2020*; *Trieu et al., 2022*; *Reynolds and Berridge, 2002*; *Castro and Bruchas, 2019*; *Schindler et al., 2012*; *Castro and Berridge, 2014*). Our aim was to measure NAcSh peptide release in response to aversive, stressful stimuli. We use custom-made microdialysis probes, intentionally modified so they are similar in size to commonly used fiber photometry probes, thus causing comparable levels of tissue damage (*Figure 1B*). Importantly, the membrane that sits above the brain region of interest is smaller in diameter when compared to a photometry probe (*Figure 1B*). After recovery, we begin microdialysis collection every 13 min, circulating artificial cerebrospinal fluid (aCSF) and collecting interstitial fluid (ISF) at a flow rate of 0.8 µL/min. After each sample collection, we add a consistent known concentration of isotopically labeled internal standard of Met-Enk and Leu-Enk of 40 amol/sample

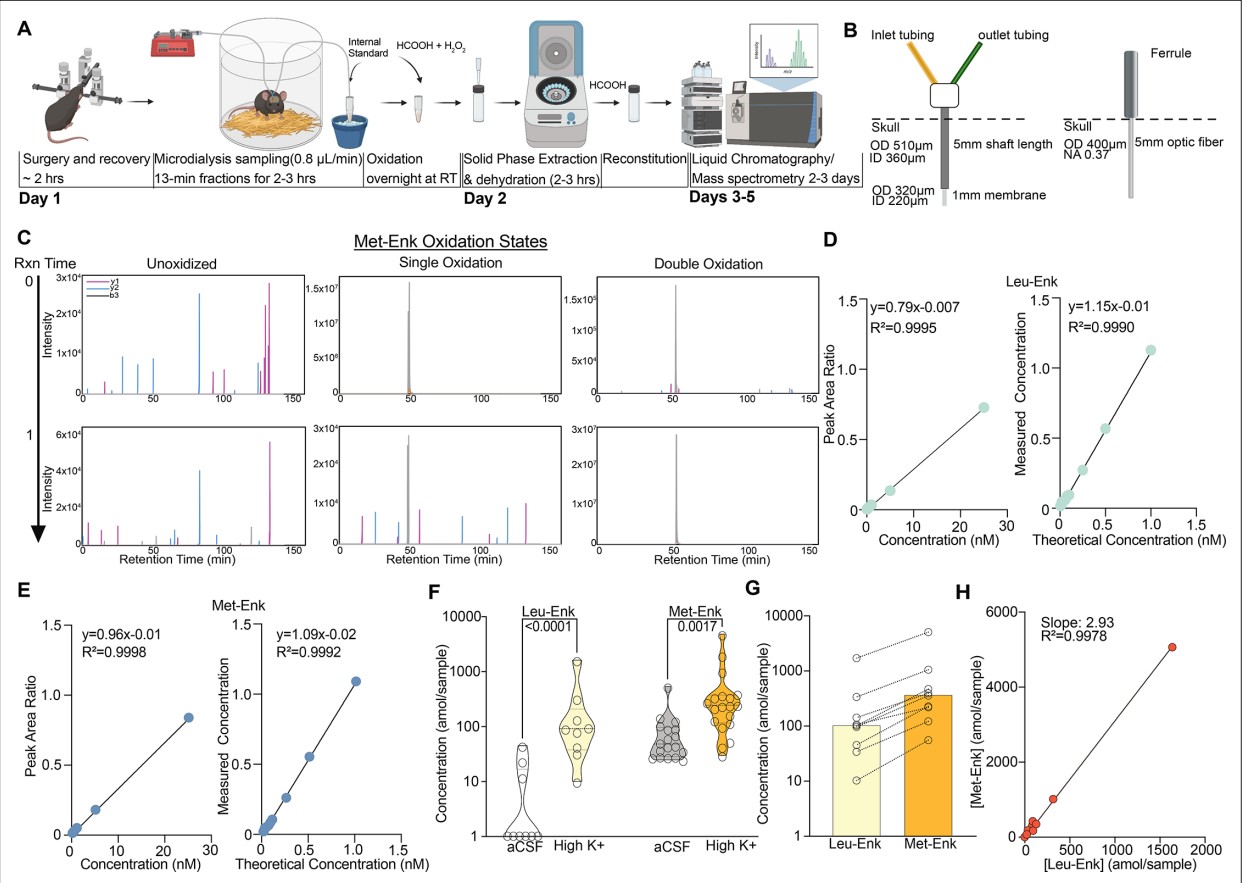

**Figure 1.** An optimized approach for in vivo Met- and Leu-enkephalin (Met- and Leu-Enk) measurement. (**A**) Timeline of in vivo sample collection on day 1 and methionine oxidation reaction overnight, sample processing on day 2, and data acquisition on the liquid chromatography/mass spectrometry (LC-MS), days 3–5. The microdialysis probe is implanted via stereotaxic surgery in the nucleus accumbens shell. The mouse is allowed to recover before being connected to the microdialysis lines. Samples are then collected at a rate of 0.8 µL/min for 13 min each. After collection is completed, the samples are oxidized overnight. On day 2, the samples undergo solid-phase extraction (SPE) and are then dehydrated and reconstituted using formic acid (HCOOH) before being acquired on the LC-MS. Panel A created with BioRender.com. (**B**) Custom microdialysis probe specifications including membrane size and inner and outer diameters (ID and OD, respectively) compared to fiber photometry probe specifications including OD of optic fiber and numerical aperture (NA). (**C**) Before the methionine oxidation at reaction (Rxn) time 0, Met-Enk exists in three different forms with varying intensities, unoxidized (multi-peak), singly oxidized (multi-peak), and doubly oxidized (single peak). After the reaction completes (Rxn time 1), most of the detected signal is in the doubly oxidized form and shows a single peak (>99% signal intensity). Y1, y2, b3 refer to the different elution fragments resulting from Met-Enk during LC-MS. (**D**) (Left) Forward calibration curve of Leu-Enk and Met-Enk showing the peak area ratios as the light standard levels are varied. (Right) Reverse calibration curve of Leu-Enk showing the relationship between the heavy standard concentration applied and the measured concentration based on the instrument. (**E**) Same setup as (**D**) but for Met-Enk. (**F**) Violin plots showing that high $K^+$ Ringer's solution increases the release of both Leu-Enk and Met-Enk compared to baseline levels in artificial cerebrospinal fluid (aCSF) (Leu-Enk n=9, Met-Enk n=18). The dashed center line indicates the median. (**G**) The evoked concentrations of Met-Enk to Leu-Enk in the same samples show that Met-Enk is consistently released at a higher level than Leu-Enk (n=9). (**H**) Met-Enk is released at a factor of 2.97 that of Leu-Enk as shown by linear regression analysis of the data in (f), suggesting a linear relationship between the two peptides. Data in (**F-H**) are transformed to a log scale. In (**F**), two-way ANOVA shows a main effect of peptide ($F_{(1, 51)}=32.30$, $p<0.0001$), solution ($F_{(1, 51)}=50.04$, $p<0.0001$), and interaction ($F_{(1, 51)}=9.119$, $p=0.0039$). The p-values reported were calculated using a Šídák's multiple comparisons test. Mean difference between baseline Leu-Enk and high $K^+$ 1.533, mean difference between baseline Met-Enk and high $K^+$ 0.6157, 95% confidence interval 0.3074–1.527 ($\log_{10}$ values).

The online version of this article includes the following source data and figure supplement(s) for figure 1:

**Source data 1.** Raw data for *Figure 1*.

**Figure supplement 1.** Technical advancements enabled using internal standards for Met- and Leu-enkephalin (Met- and Leu-Enk) detection.

**Figure supplement 1—source data 1.** Raw data for *Figure 1—figure supplement 1*.

to the collected ISF for the accurate identification and quantification of endogenous peptide. These internal standards have a different mass/charge (m/z) ratio than endogenous Met- and Leu-Enk. Thus, we can identify true endogenous signal for Met-Enk and Leu-Enk (*Figure 1—figure supplement 1A and C*) versus noise, interfering signals, and standard signal (*Figure 1—figure supplement 1B and D*). Importantly, we have resolved one of the main barriers preventing the accurate quantification of Met-Enk: the variable oxidation of the methionine residue during sample processing, which results in the signal for Met-Enk being split among multiple m/z values, thereby impairing sensitivity. In order to ensure that the signal for Met-Enk is represented by a single m/z value, we converted all methionyl residues to the doubly oxidized sulfone prior to nLC-MS analysis, which allows us to reliably quantify Met-Enk and differentiate it from Leu-Enk without compromising the detection of either peptide (*Pesavento et al., 2007*). Beyond opioid peptides, this approach represents a key advance and can be applied to any peptide containing a methionine residue. As seen in *Figure 1C*, before the methionine oxidation reaction at time 0, Met-Enk can be observed in unoxidized, and singly and doubly oxidized forms with varying intensities. However, after the reaction is complete at time 1, Met-Enk is observed as doubly oxidized with much higher intensity, enabling accurate quantification.

To increase the signal-to-noise ratio, we also performed solid-phase extraction after the oxidation reaction, which removes any salt from the collected samples. To determine the lower limit of quantification (LLOQ), forward and reverse curves were created using varying concentrations of the Met- and Leu-Enk standards in accordance with the Clinical Proteomics Tumor Analysis Consortium (CPTAC) guidelines (*Carr et al., 2014*). We determined that the LLOQ for both Leu-Enk and Met-Enk is 40 amol/sample (*Figure 1D and E*). The improved quantification limit means increased temporal resolution. We now need less volume (10 µL or less) and shorter sampling times: 10 min or less, rather than 15–20 min as compared to previous studies (*Al-Hasani et al., 2018*; *DiFeliceantonio et al., 2012*). To test the utility of the method in measuring real-time changes in peptide levels in vivo, we evoked the release of the peptides using high $K^+$ Ringer's solution, which depolarizes the cells and causes the release of dense core vesicles containing neuropeptides (*Ripley et al., 1997*). This depolarization state significantly increased the release of Met- and Leu-Enk in the NAcSh as compared to baseline (*Figure 1F*). Interestingly, Leu-Enk showed a greater fold change compared to baseline than did Met-Enk, with the fold changes being 28 and 7, respectively, based on the data in *Figure 1F*. Previous studies reporting opioid peptide measurements rarely represent raw concentrations measured. Instead, most past studies reported changes in peptide concentrations as % of baseline (*Mabrouk et al., 2011*; *Al-Hasani et al., 2018*; *DiFeliceantonio et al., 2012*). One key advantage of our improved method is that we can detect reliable measurements of both Met- and Leu-Enk at baseline reported as concentrations in the amol/sample range. We then quantified the ratio of Met- to Leu-Enk after applying high $K^+$ solution and showed that this ratio is 3:1 in the NAcSh (*Figure 1G and H*). The release of Met-Enk is postulated both in vitro and ex vivo to be between three and four times that of Leu-Enk (*Comb et al., 1982*; *Hughes et al., 1997*; *Henderson et al., 1978*). We are the first to not only corroborate the predicted ratio in vivo in freely behaving animals, but specifically in the NAcSh. Interestingly, this 3:1 ratio is almost identical to that previously reported in homogenized human tissue from the sympathoadrenal systems using radioimmunoassays (*Yoshimasa et al., 1982*). Importantly, this conserved ratio highlights the usefulness of rodent models for investigating the highly conserved opioid system and demonstrates its potential in translation to humans.

## Experimenter handling drives the release of Met- and Leu-Enk in the NAcSh

To highlight the applicability and value of this improved approach, we investigated the changes in the levels of enkephalins following acute stress. It has been hypothesized that enkephalins act as anti-stress signals based on studies that have shown increased preproenkephalin transcript levels following acute stress (*Mansi et al., 2000*; *Dumont et al., 2000*; *Ceccatelli and Orazzo, 1993*; *Valentino and Van Bockstaele, 2015*). These studies have included regions such as the locus coeruleus, the ventral medulla, the basolateral nucleus of the amygdala, and the nucleus accumbens core and shell. Studies using global knockout of enkephalins have shown varying responses to chronic stress interventions, where male knockout mice showed resistance to chronic mild stress in one study, while another study showed that enkephalin-knockout mice showed delayed termination of corticosterone release (*Melo et al., 2014*; *Bilkei-Gorzo et al., 2008*). Moreover, decreased enkephalin expression in the NAc was

correlated with an increase in the susceptibility to social defeat (*Nam et al., 2019*). This body of literature supports the hypothesis that enkephalins participate in stress encoding and reactivity. We were specifically interested in how enkephalins mediate acute stress responses. Our group has also recently demonstrated that the photostimulation of a subpopulation of peptidergic neurons in the NAcSh drives aversion (*Al-Hasani et al., 2015*), which further suggests a role for neurons in the NAcSh during aversive stimuli such as stress. Despite the wealth of data implicating these peptides in the stress response, it has not been possible to measure the dynamics of real-time peptide release in response to acute stressful stimuli. We sought to test the hypothesis that Met- and Leu-Enk are released in the NAcSh following acute stress. Using the nLC-MS detection method, we successfully monitored dynamic changes of Met- and Leu-Enk during two different stressors. The first stressor was defined as experimenter handling, which involves the experimenter scruffing the mouse while connecting to the dialysis setup (*Figure 1A*). The second stressor was exposure to predator odor (i.e. fox urine).

We show that both Met- and Leu-Enk are released following experimenter handling, and that the levels decrease within minutes after exposure to the stress (*Figure 2A*). We also analyzed the data with sex as a main effect and did not find any differences between males and females after correcting for multiple comparisons (*Figure 2—figure supplement 1A*). We corroborate our findings from *Figure 1G and H* showing that the ratio of 3:1 Met-Enk to Leu-Enk is conserved following experimenter handling (*Figure 2C*). Another key finding is that peptide release appears to be precisely controlled. We show that the amol/sample of Met-Enk released at the beginning of the experiment is predictive of later release. For example, a higher level released (in the fmol/sample range) during the first collections results in lower levels released (in the amol/sample range) during later collections and vice versa (*Figure 2D*). We show a negative correlation (~−0.3) between the first two samples (~26 min) and the later time points, samples 3–10 (~104 min). Importantly, this relationship is not due to Met-Enk depletion as the circulation of high $K^+$ aCSF is sufficient to drive the release of Met-Enk at the end of an experiment and reverse the negative correlation (*Figure 2E*).

To test whether our findings regarding peptide release correlated with neuronal activity following the same stressor, we also monitored enkephalinergic neuron activity using fiber photometry following experimenter handling. To do this, we stereotaxically injected a Cre-dependent adeno-associated virus containing the calcium sensor GCaMP6s in the NAcSh of preproEnkephalin-Cre (*Penk*-Cre) mice and implanted fiber photometry probes (*Figure 2F*). During handling, we observed increased calcium activity in enkephalinergic neurons, even after it was repeated twice within a few minutes (*Figure 2G*). Individual responses to experimenter handling varied appreciably, as shown by the heatmaps and bar plots (*Figure 2H and I*); such variation could be due to slight differences in probe placement (*Figure 3—figure supplement 1B and D*) as peptidergic neurons in the dNAcSh encode reward while neurons in the vNAcSh encode aversion (*Al-Hasani et al., 2015*; *de Jong et al., 2019*). Furthermore, the enkephalin precursor is expressed across the dorsal-ventral axis as shown in the in situ hybridization atlas (Allen Brain Institute). The fiber photometry data and peptide release data are concordant, as enkephalinergic neurons are activated following experimenter handling, which drives Met- and Leu-Enk somatodendritic release. However, the photometry and peptide release data differ in that repeated neuronal activation may not necessarily lead to repeated peptide release, as suggested by the decrease in peptide release over time in *Figure 2D*. Additionally, fiber photometry cannot provide information about Met- and Leu-Enk independently, as it relies on the precursor gene, *Penk*. Taken together, our data highlight the value and importance of coupling fiber photometry results with in vivo measurements of neuropeptides to determine release dynamics.

## Predator odor causes the release of Met-Enk in the NAcSh

We wanted to test another stressor that is similar in nature to experimenter handling, as the experimenter represents aversive visual, olfactory, and touch stimuli to the mouse. Fox odor has been widely demonstrated to be aversive to rodents and other mammalian species (*Apfelbach et al., 2005*; *Endres and Fendt, 2009*; *Farmer-Dougan et al., 2005*). The benefit of using predator odor is that it represents what mice would encounter in the wild, outside of the lab environment. It also allows us to probe conserved circuits in laboratory mice related to aversion, threat, and defensive behaviors. Prior reports have shown that predator odor leads to an increase in the expression of the neuronal activation-associated transcription factor fos in enkephalin-positive neurons in different brain regions, including the NAcSh (*Hebb et al., 2004*; *Asok et al., 2013*). Therefore, we sought to

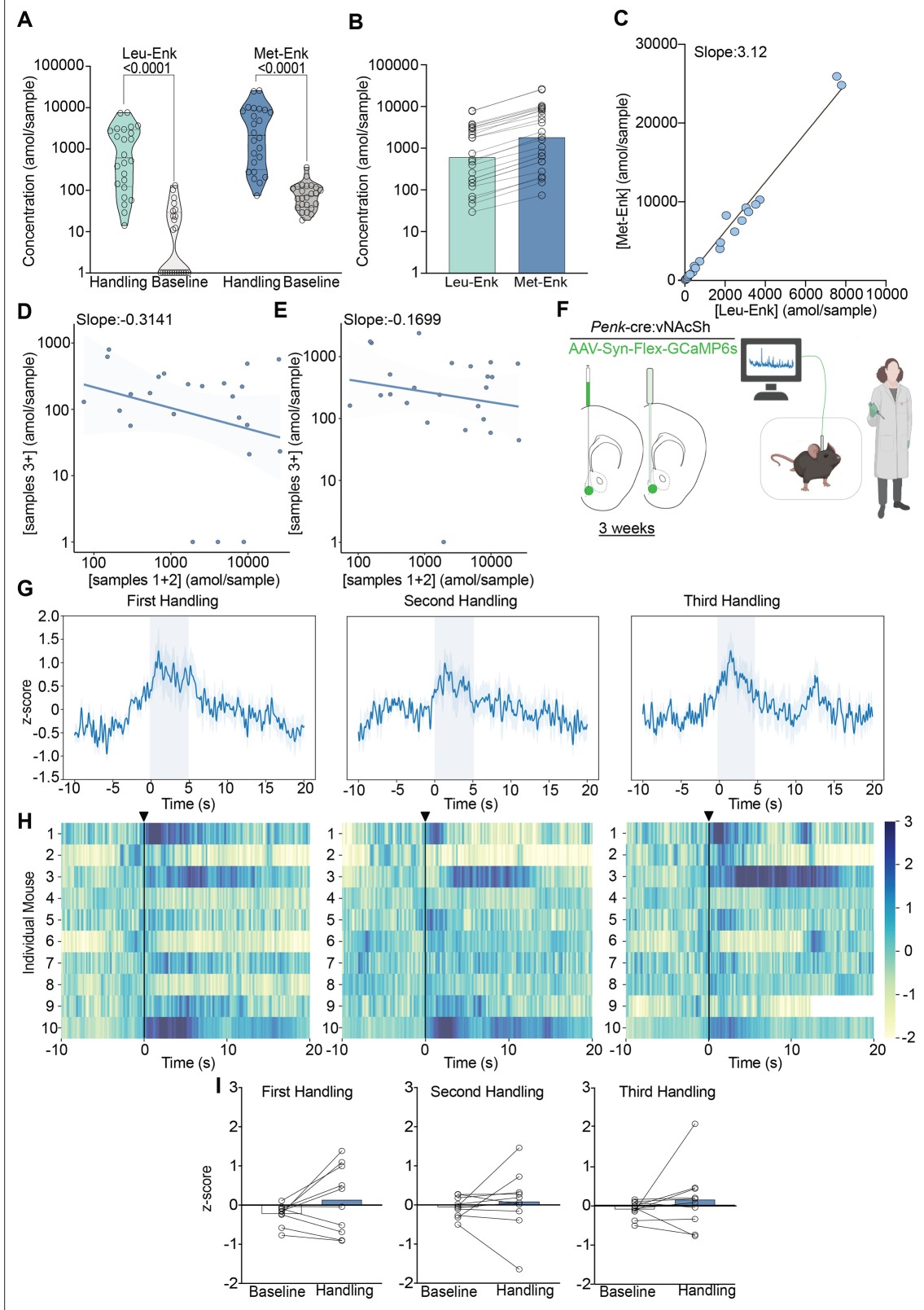

**Figure 2.** Experimenter handling evokes the release of Met- and Leu-enkephalin (Met- and Leu-Enk) in the nucleus accumbens shell (NAcSh). (**A**) Experimenter handling during microdialysis causes a significant increase in the release of Leu- and Met-Enk in comparison to baseline. The two-way ANOVA on log-transformed data showing a significant effect of peptide F(1,92)=32.58, p<0.0001, handling F(1,92)=146.1, p<0.0001, and interaction F(1,92)=4.778, p=0.0314. Šídák's multiple comparisons test was conducted, and p-values are shown on the figure. Mean difference between baseline

*Figure 2 continued on next page*

*Figure 2 continued*

Leu-Enk and handling 2.055, mean difference between baseline Met-Enk and handling 1.426, 95% confidence interval 0.05751–1.201 ($\log_{10}$ values). (**B**) During experimenter handling, Met-Enk is consistently released at a higher rate than Leu-Enk in the same samples. (**C**) Linear regression analysis of the data in (**B**) shows that Met-Enk is released at a rate of 3.12 times the rate of Leu-Enk during experimenter handling, suggesting a linear relationship between the two peptides. (**D**) A negative correlation (–0.3141) shows that if a high concentration of Met-Enk is released in the first two samples, the concentration released in later samples is affected; such influence suggests that there is regulation of the maximum amount of peptide to be released in NAcSh. (**E**) The negative correlation in panel (**D**) is reversed by using high $K^+$ buffer with a negative correlation coefficient of –0.1699 to evoke Met-Enk release, suggesting that the limited release observed in (**D**) is due to modulation of peptide release rather than depletion of reserves. Data in (**A–E**) are transformed to a log scale, and n=24 animals. In panel (**A**), two-way ANOVA shows a main effect of peptide, treatment, and interaction. The p-values reported were calculated using a Šídák's multiple comparisons test. (**F**) Schematic describing the viral strategy and probe placement for the fiber photometry experiment in *Penk*-Cre mice injected with the calcium sensor GCaMP6s in the NAcSh. Panel F created with BioRender.com. (**G**) Average z-score trace following the first, second, and third events involving experimenter handling. (**H**) Heatmaps showing individual mouse z-scored fiber photometry responses before and after experimenter handling. (**I**) Bar graphs showing the averaged z-score responses before experimenter handling and after experimenter handling (n = 10 for **G–I**).

The online version of this article includes the following source data and figure supplement(s) for figure 2:

**Source data 1.** Raw data for *Figure 2*.

**Figure supplement 1.** There are no significant differences between male and female responses to handling or fox odor exposure.

**Figure supplement 1—source data 1.** Raw data for *Figure 2—figure supplement 1*.

determine whether enkephalins are released in the NAcSh following exposure to fox urine. We first measured calcium activity of enkephalinergic neurons using fiber photometry after exposure to fox urine (*Figure 3A*). We show that upon first exposure, the activity of enkephalinergic neurons is markedly increased (*Figure 3B–D*). However, this enkephalinergic activity is significantly attenuated during second and third exposures that are separated by 5 min, suggesting habituation to the stimulus (*Figure 3B–D*). To determine whether the response to fox urine may also be due to the novelty of the weigh boat used to introduce the fox urine, we measured enkephalinergic calcium activity after introducing a weigh boat containing water to the home cage (*Figure 3—figure supplement 2A*). Interestingly, we show that enkephalinergic neurons also respond to the introduction of a novel object, and the response to the water-containing weigh boat is attenuated upon repeated exposure, which is what we observed for fox urine (*Figure 3—figure supplement 2B–D*). We compared the enkephalinergic neuron activation in response to both water and fox urine for each individual mouse that experienced both, and we see similar calcium transient responses to both the introduction of a novel object and the odor of a predator, suggesting the same subpopulation of enkephalinergic neurons may have been activated (*Figure 3—figure supplement 2E*). Interestingly and importantly, only recently have researchers shown that a higher level of spiking in the striatum is not always reflected in greater photometry transients (*Legaria et al., 2022*). In the future, it will be important to determine whether the fox urine response indeed shows higher action potential spiking than the weigh boat with water. Although the calcium responses to both the fox urine and water-containing weigh boats were similar in magnitude, it may be that different subpopulations of enkephalinergic neurons are activated in the NAcSh. Further exploration of the specific neurons that respond to the introduction of a novel object or the odor of a predator is warranted. We concluded that enkephalinergic neurons are activated by fox odor using fiber photometry. We were then interested in testing whether activation of enkephalinergic neurons leads to the release of Met- and Leu-Enk, so we measured peptide levels in the NAcSh after fox odor using our method of microdialysis and nLC-MS (*Figure 3E*).

In this separate group of animals, we show that there is a significant increase in Met-Enk release after exposure to fox odor (*Figure 3F*). The response to predator odor varies appreciably: the responses are almost evenly split between elevated release, release with no elevation, or no detectable release (*Figure 3G*). Enkephalins are thought to mediate resilience to stress, so the individual differences in responses to predator odor may indicate differences in stress adaptability (*Valentino and Van Bockstaele, 2015*; *Melo et al., 2014*; *Bérubé et al., 2014*; *Bérubé et al., 2013*). Moreover, no sex differences in the responses to fox urine were observed, similar to responses to experimenter handling (*Figure 2—figure supplement 1*). It is also worth noting that we only observed Leu-Enk release in 3 of 16 experiments. This is likely due to Leu-Enk not being released following predator odor or because it is released at a level below the LLOQ, as suggested by the observed 3:1 ratio of Met-Enk to Leu-Enk (*Figure 3G*).

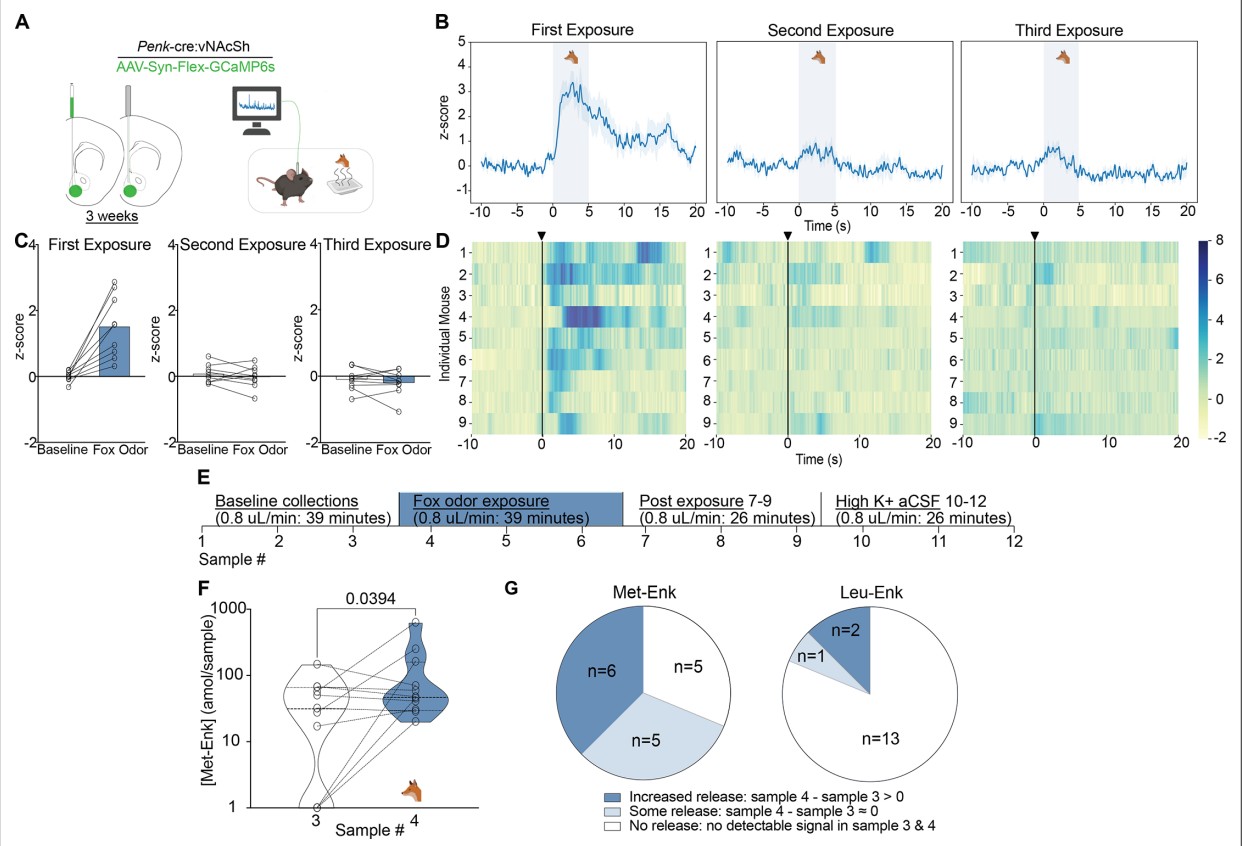

**Figure 3.** Fox odor exposure activates enkephalinergic neurons and drives the release of Met-enkephalin (Met-Enk) in the nucleus accumbens shell (NAcSh). (**A**) Viral strategy and probe placement for the fiber photometry experiment in *Penk*-Cre mice injected with the calcium sensor GCaMP6s in the NAcSh. (**B**) Averaged z-score traces of the first, second, and third exposure to fox odor. (**C**) Bar graphs showing the averaged z-scored fiber photometry responses before and after exposure to fox odor. (**D**) Heatmaps showing individual mouse z-scored fiber photometry responses before and after exposure to fox odor (n=9, for **A–D**). (**E**) Experimental timeline describing the fox odor microdialysis experiment. (**F**) There is a significant increase in Met-Enk concentration during fox odor exposure (sample 4), in comparison to sample 3 before odor exposure. Data is transformed to a log scale, and the p-value was calculated using a two-tailed paired t-test (n=11), t=2.369, df = 10, mean difference 0.1388, 95% confidence interval –0.5419 to 0.2644. (**G**) Pie charts showing the variation across animals in Met-Enk and Leu-Enk release profiles in response to exposure to fox odor, suggesting that such exposure may selectively cause the release of Met-Enk but not Leu-Enk. Increased release showed higher concentration of Met-Enk during exposure to fox odor (sample 4) than the sample before (sample 3). Some releases showed a comparable level to the sample before exposure, and no release showed no quantifiable concentration during exposure to fox odor. Panel A, B and F created with BioRender.com.

The online version of this article includes the following source data and figure supplement(s) for figure 3:

**Source data 1.** Raw data for *Figure 3*.

**Figure supplement 1.** Hit map of microdialysis and fiber photometry probes.

**Figure supplement 2.** Water exposure activates enkephalinergic neurons in the nucleus accumbens shell (NAcSh).

**Figure supplement 2—source data 1.** Raw data for *Figure 3—figure supplement 2*.

## Discussion

In summary, this work allows for multiple unique advances in studying endogenous opioids and neuro-peptides in general. Although there has been substantial progress in fluorescent sensor-based techniques, electrochemical approaches, and immunoaffinity-based assays, the investigation of opioid peptides has lagged due to their unique properties: sequence similarity, diversity, and low concentrations. Here, we address these challenges by advancing microdialysis coupled with nLC-MS. First, we have refined a method of methionine stabilization (*Pesavento et al., 2007*; *Rosei et al., 1991*; *Finn et al., 2004*) that is transferable to other neuropeptides and have decreased the LLOQ and temporal resolution, which are typically the main critiques of microdialysis and LC-MS approaches. As we have shown, if Met-Enk is not fully oxidized prior to LC-MS analysis, it is hard to quantify because of the

presence of differentially oxidized species. The diversity of the signal detected impedes the analysis of a single sample and may lead to inaccurate concentration values. Here, we show that the oxidation of methionine allows for the stable detection of not only Met-Enk but also the simultaneous and selective detection of Leu-Enk.

Second, we quantified the relationship between Leu- and Met-Enk and offer a definitive in vivo ratio (3:1 Met-Enk to Leu-Enk) that has long been in question; this ratio corroborates human data (*Yoshimasa et al., 1982*) and suggests that mice are a useful model system for the study of enkephalinergic release dynamics. Based on this, future studies investigating the use of opioid peptides as biomarkers for psychiatric conditions will be readily translatable from rodents to humans and will allow for ease in reverse translation efforts. There is already some demonstrated interest in measuring opioid peptide levels in humans due to their roles in mood disorders, substance use, and pain (*Al-Hakeim et al., 2020*; *Kamel et al., 2007*). Moreover, prior to our study, the predicted ratio of Met- to Leu-Enk was predicted based on the mRNA transcript sequence, so we are the first to confirm that posttranslational cleavage indeed yields the predicted ratio. We also noted that Leu-Enk showed a greater fold increase relative to baseline after depolarization with high $K^+$ buffer as compared to Met-Enk. This may be due to increased Leu-Enk packaging in dense core vesicles compared to Met-Enk or since there are two distinct precursor sources for Leu-Enk, namely both proenkephalin and prodynorphin, while Met-Enk is mostly cleaved from proenkephalin (*Conway et al., 2022*). Third, we identify a role for the NAcSh in the acute stress response by demonstrating local release of Met- and Leu-Enk after stress exposure using our improved microdialysis and nLC-MS method. Although studies have suggested a role for the NAcSh in stress through measurements of *Penk* transcripts, it had not been definitively shown that there is local release of enkephalins after stress exposure (*Nam et al., 2019*; *Hebb et al., 2004*). Our findings showed a robust increase in peptide release at the beginning of experiments, which we interpreted as due to experimenter handling stress that directly precedes microdialysis collections. However, there are other technical limitations to consider, such as the fact that we were collecting samples from mice that were recently operated on. Another consideration is that the circulation of aCSF through the probe may cause a sudden shift in oncotic and hydrostatic forces, leading to increased peptide release to the extracellular space. As such, we wanted to examine our findings using a different technique, so we chose to record calcium activity from enkephalinergic neurons – the same cell population leading to peptide release. Using fiber photometry, we showed that enkephalinergic neurons are activated by stress exposure, both experimenter handling and fox odor, thereby adding more evidence to suggest that enkephalinergic neurons are activated by stress exposure, which could explain the heightened peptide levels at the beginning of microdialysis experiments. Repeated experimenter handling led to repeated activation of enkephalinergic neurons. However, fox odor exposure showed significant attenuation after the first exposure. This is consistent with prior studies showing fast habituation to olfactory stimuli in rodents after initial exposure (*Apfelbach et al., 2005*; *Endres and Fendt, 2009*; *Farmer-Dougan et al., 2005*; *Sundberg et al., 1982*). We also showed using fiber photometry that the novelty of the introduction of a foreign object to the cage, before adding fox odor, was sufficient to activate enkephalinergic neurons. This is consistent with the object being a stressor as it is newly introduced to the home cage. Alternatively, it could also mean that the stress response is more nuanced and is impacted by the novelty of the stressor. This is an area of interest to our group, and we will design future experiments to disentangle the novelty from the stressor in order to understand the enkephalinergic system better. Given that previous studies implicating enkephalins in stress used mRNA transcript measurements of proenkephalin, it was not possible to determine whether Leu- and/or Met-Enk were involved in the acute stress response. Our study suggests that both peptides are involved in experimenter handling stress; however, this is less clear following exposure to fox odor, where Met-Enk appears to be primarily involved. Most of our experiments with fox odor do not show Leu-Enk release; however, that could be due to Leu-Enk levels being below the LLOQ since we have demonstrated that it is three times lower than Met-Enk. Fourth, we show that Met- and Leu-Enk are released during acute stress in the NAcSh, offering evidence for their role in gating the stress response, which was suggested based on studies of enkephalins in the locus coeruleus (*Valentino and Van Bockstaele, 2015*). Therefore, combined with other studies, our work suggests that the role of enkephalins in stress is generalizable to other brain regions, namely the ventral striatum. Based on our findings, it remains unclear how enkephalins modulate the stress response. It will be interesting to investigate whether enkephalinergic signaling modulates stress

coping behaviors. Future studies will focus on determining the effect of the local release of Met- and Leu-Enk in the NAcSh on responses to stress in rodents.

Finally, when a limited amount of peptide is released over time, we show that peptide release is biologically constrained and can be experimentally driven using high $K^+$ buffer or photo- or chemogenetic stimulation (*Al-Hasani et al., 2018*). We also show that data from fiber photometry corroborate peptide release data; however, fiber photometry does not offer the necessary specificity to correlate activation with release, or the distinction between different peptides that arise from the same precursor molecule. Additionally, it is important to stress that our fiber photometry studies used a calcium sensor as a proxy for neuronal activity, and the temporal scale was on the order of seconds. A recent study using an opioid-peptide fluorescent sensor shows temporal scales on the order of minutes, which mirrors microdialysis timelines for neuropeptide detection and offers evidence that neuropeptide dynamics are generally much slower than neurotransmitters in vivo (*Abraham et al., 2021*). In conclusion, this improved peptide detection method enables biological advances through an optimized approach for Met- and Leu-Enk detection in vivo with broad applicability to other neuropeptides.

# Materials and methods

## Key resources table

| Reagent type (species) or resource | Designation | Source or reference | Identifiers | Additional information |
|---|---|---|---|---|
| Strain, strain background (C57BL/6 mice) | C57BL/6 | The Jackson laboratory | Strain #:000664 (Jackson laboratory) RRID:MGI:2159769 | Ongoing breeding in laboratory |
| Strain, strain background (proproenkephalin-Cre, C57BL/6 background, mice) | *Penk*-Cre | Moron-Concepcion Lab (PMID:34646022) | | |
| Strain, strain background (AAV5.Syn.Flex.GCaMP6s. WPRE.SV40) | GCaMP6s | Addgene | viral prep # 100845-AAV5 RRID:Addgene_100845 | pAAV.Syn.Flex.GCaMP6s.WPRE.SV40 was a gift from Douglas Kim and GENIE Project (Addgene viral prep # 100845; http://n2t.net/addgene: 100845) |
| Software | Guppy | PMID:34930955 | RRID:SCR_022353 | |
| Software | Python, Matplotlib | https://matplotlib.org/ | RRID:SCR_008394 | |
| Software | GraphPad Prism | https://www.graphpad.com/ | RRID:SCR_002798 | |
| Software | Affinity Designer | https://affinity.serif.com/en-gb/ | RRID:SCR_016952 | |
| Antibody | guinea pig anti-GFAP | Synaptic systems | RRID:AB_10641162 | |
| Antibody | Alexa Fluor 594 | Invitrogen | RRID:AB_141930 | |
| Chemical compound, drug | Vectashield with DAPI | Vector labs | RRID:AB_2336788 | |
| Chemical compound, drug | NeuroTrace 530/615 Red Fluorescent Nissl Stain – Solution in DMSO | Invitrogen | Catalog # N21482 RRID:AB_2620170 | |
| Chemical compound, drug | Vectashield | Vector labs | RRID:AB_2336787 | |
| Peptide, recombinant protein (isotope labeled Leu-Enkephalin, YGGF**L**) | Leu-Enkephalin (heavy isotope labeled) | New England Peptide | | Labeled L-Leucine ($^{13}C_6$, $^{15}N$) at the C terminus of the peptide |
| Peptide, recombinant protein (isotope labeled Met-Enkephalin, YGGF**M**) | Met-Enkephalin (heavy isotope labeled) | New England Peptide | | Labeled L-Phenylalanine ($^{13}C_9$, $^{15}N$) was added |
| Peptide, recombinant protein (Leu-Enkephalin, YGGFL) | Leu-Enkephalin (standard) | New England Peptide | | Non-isotope labeled standard |
| Peptide, recombinant protein (Met-Enkephalin, YGGFM) | Met-Enkephalin (standard) | New England Peptide | | Non-isotope labeled standard |
| Commercial assay or kit | Parkell C&B Metabond kit | Optimus dental supply | SKU 04-PKL-S380 | |

*Continued on next page*

*Continued*

| Reagent type (species) or resource | Designation | Source or reference | Identifiers | Additional information |
|---|---|---|---|---|
| Chemical compound, drug | Acetonitrile (MeCN) | JT Baker | Cat # 9829-03 | |
| Chemical compound, drug | Water | JT Baker | Cat # 4218-03 | |
| Chemical compound, drug | Formic acid | Sigma-Aldrich | Cat # 56302 | |
| Chemical compound, drug | Methanol (MeOH) | Fluka | Cat # 34966 | |
| Chemical compound, drug | Hydrogen peroxide | Fisher Chemical | Cat #H325-100 | |
| Chemical compound, drug | NaCl | Sigma-Aldrich | | |
| Chemical compound, drug | KCl | Sigma-Aldrich | | |
| Chemical compound, drug | $CaCl_2$ | Sigma-Aldrich | | |
| Chemical compound, drug | $MgCl_2$ | Sigma-Aldrich | | |
| Chemical compound, drug | $NaH_2PO_4$ | Sigma-Aldrich | | |
| Chemical compound, drug | $NaHCO_3$ | Sigma-Aldrich | | |
| Chemical compound, drug | HEPES | Sigma-Aldrich | | |
| Chemical compound, drug | Glucose | Sigma-Aldrich | | |
| Chemical compound, drug | Fox urine | PredatorPee fox pee brand | | https://predatorpeestore.com/ |
| Chemical compound, drug | Ketamine | | | |
| Chemical compound, drug | Xylazine | Dechra | Rompun | |
| Chemical compound, drug | Acepromazine | Boehringer Ingelheim Animal Health USA Inc | NDC 0010-3827-01 | |
| Chemical compound, drug | Paraformaldehyde | Sigma-Aldrich | | |
| Chemical compound, drug | Triton X-100 | Sigma-Aldrich | | |
| Chemical compound, drug | Goat serum | Sigma-Aldrich | | |
| Chemical compound, drug | PBS | Sigma-Aldrich | | |
| Other | Probes and setup | BASi | | Microdialysis probes used in this study are no longer carried by BASi, detailed information in the sections below. |
| Other | Probes and setup | TDT, Doric lenses | | |
| Other | Laboratory pipetting needles with 90° blunt ends, 16-gauge, 2-inch length | Cadence Science, | Cat # 7938 | |
| Other | C18 extraction disks, diam.=47 mm, 20 pack; | Empore | Cat# 66,883U | |
| Other | Adapter for stage tipping | Glygen | Cat# CEN.24 | |
| Other | 1.5 mL tubes | Axygen | Cat# MCT-175-C | |
| Other | Eppendorf centrifuge | Eppendorf, | model 5424R | |
| Other | Vortex | Labnet VX100 | | |
| Other | Autosampler vials | Sun-Sri | Cat # 200046 | |
| Other | Autosampler vial caps | Sun-Sri | Cat # 501 382 | |
| Other | EASY nLC 1000 | Thermo Scientific | RRID:SCR_014993 | |

*Continued on next page*

Continued

| Reagent type (species) or resource | Designation | Source or reference | Identifiers | Additional information |
|---|---|---|---|---|
| Other | Q-Exactive Plus Hybrid Quadrupole Orbitrap | Thermo Scientific | RRID:SCR_020556 | |
| Other | EASY-Spray column, 75 µm × 50 cm PepMap | Thermo Scientific | Cat # ES903 | |

## Animals

Adult male and female C57BL/6 mice at 8–15 weeks of age were used for microdialysis procedures. For fiber photometry, adult preproEnkephalin-Cre (*Penk*-Cre) (*Castro et al., 2021*) male and female mice at ages between 15 and 24 weeks of age were used to measure enkephalinergic cell activity. Mice were group-housed and kept at a 12 hr light/dark cycle with ad libitum access to water and food. The holding facility was both temperature- and humidity-controlled. All surgical and behavioral procedures were approved by the Washington University School of Medicine's animal care guidelines through protocols approved by the Institutional Animal Care and Use Committee (Protocol# 23-0261) in accordance with federal animal use regulations. The animals were cared for by lab staff as well as the Department of Comparative Medicine at Washington University School of Medicine.

## Chemicals

Met- and Leu-Enk standards, including isotopically labeled forms, were synthesized at New England Peptide and can be ordered by the research community. The heavy isotope-labeled Leu-Enk is synthesized with a fully labeled L-Leucine ($^{13}C_6$, $^{15}N$) at the C terminus of the peptide (YGGF**L**). For Met-Enk, a fully labeled L-Phenylalanine ($^{13}C_9$, $^{15}N$) was added (YGG**F**M). The resulting mass shift between the endogenous (light) and heavy isotope-labeled peptides is 7 Da and 10 Da, respectively. Heavy isotope amino acids are guaranteed >99% pure. Peptide purity specifications are set at ≥95% pure by area on a regular HPLC gradient with UV detection. aCSF contained 124 mM NaCl, 2.5 mM KCl, 2 mM CaCl$_2$, 1 mM MgCl$_2$, 1.25 mM NaH$_2$PO$_4$, 24 mM NaHCO$_3$, 5 mM HEPES, 12.5 mM glucose adjusted to pH 7.4 with NaOH. High K$^+$ Ringer's solution contained 48 mM NaCl, 100 mM KCl, 2.4 mM CaCl$_2$, 0.85 mM MgCl$_2$ adjusted to pH 7.4 with NaOH as described in *Al-Hasani et al., 2018*. All salt compounds were purchased from Sigma (St. Louis, MO, USA).

## In vivo microdialysis

Custom microdialysis probes (BASI Inc part MD-2206) of 5 mm shaft length and 1 mm polyacrylonitrile membrane with a 30 kDa molecular weight cutoff were used for the in vivo experiments. The probes were flushed prior to surgical implantation for 15 min at 10 µL/min with aCSF as directed by the manufacturer using a BASI syringe pump (MD 1001). For the surgical procedure, mice were briefly anesthetized in an induction chamber at 3% isoflurane before being placed in a stereotaxic frame (Kopf) secured with ear and bite bars. During the surgery, the isoflurane level was kept at 1.5–2%. The probe was then inserted into the ventral NAcSh (stereotaxic coordinates from bregma: +1.3 [AP], ±0.5 [ML], −5.0 mm [DV]). Implanted probes were secured with dental adhesive (C&B Metabond kit) followed by cyanoacrylate (Loctite Super Glue Gel Control).

After the completion of the surgery, each mouse was allowed to recover for 30 min. The mouse was then connected to an inlet perfusion line to circulate aCSF or high K$^+$ Ringer's solution and an outlet perfusion line to collect the ISF from the NAcSh. The outlet line directly collected sample in a tube that is kept at 4°C throughout the experiment. The flow of aCSF was then turned on, and sample collection time started as the sample began to collect in the outlet tube. The flow rate of aCSF was set to 0.8 µL/min during sample collection. Fractions were collected every 13 min, resulting in a 10 µL sample. After each collection, the samples were spiked with 2 µL of 12.5 fmol isotopically labeled Met-Enk and Leu-Enk. The microdialysis rig (BASI Raturn MD 1404) was equipped with bedding, regular mouse chow, and water ad libitum. While connected to the perfusion lines, the mice were allowed to move freely. For the experiments measuring evoked release, the high K$^+$ Ringer's solution was perfused through the lines instead of aCSF at 0.8 µL/min, and fractions were collected every 13 min as described previously. The samples were similarly spiked with 2 µL of 12.5 fmol isotopically labeled Met-Enk and Leu-Enk.

For the experiments where the animals were exposed to the fox urine odor, 3 mL of fox urine was absorbed into a kim wipe contained in a chemical weigh boat. The weigh boats were prepared and remained in a fume hood for the experiment to prevent the odor permeating the experimental space before the planned collection. Then, the experimenter placed the weigh boat in the microdialysis rig and allowed the mouse to explore it. The exposure was standardized to span three fractions (39 min) during each of the experiments while aCSF was being perfused through the lines.

## In vivo sample processing
### List of abbreviations

FA, formic acid
MeCN, acetonitrile
MeOH, methanol
DIA, data-independent acquisition
AGC, automatic gain control nLC-MS, capillary liquid chromatography interfaced to a mass spectrometer
MS1, mass spectra of peptide precursors
MS2, fragmentation mass spectrum of peptide from precursor ion

### Equipment
#### Methionine oxidation reaction

For Met-Enk detection and stabilization, the animal samples underwent a methionine oxidation reaction (*Pesavento et al., 2007*). To each sample vial, 28 µL of 0.1% (vol/vol) formic acid (FA) was added, and then the samples were transferred to 0.5 mL tubes from the collection tubes. Afterward, 40 µL of a mixture of 6% FA (vol/vol) and 6% hydrogen peroxide (vol/vol) was added to each of the sample tubes and mixed and gently spun. The samples were then incubated overnight at room temperature.

#### Solid-phase extraction with C18 stage tips

After the methionine oxidation reaction, a solid-phase extraction process was performed using C18 stage tips. Stage tips were conditioned before sample loading. To each of the stage tips, 100 µL of methanol is added and then spun for 2 min at 2000 rpm in microcentrifuge tubes. The stage tips were then rinsed with 100 µL of a mixture of 50% acetonitrile (MeCN; vol/vol) and 0.1% FA (vol/vol) followed by a 2 min spin at 2000 rpm. This was followed by a rinse in 0.1% FA (vol/vol) and a 2 min spin at 3000 rpm. The final rinse was done by adding 100 µL of 1% FA (vol/vol) to each of the stage tips. Following the final rinse, the samples were added to the C18 stage tips and spun for 10 min at 0.8 rpm. After ensuring that the samples passed through the stage tips, the samples were desalted with 0.1% FA (vol/vol) and spun for 1.5 min at 3000 rpm twice. After the spins were done, the stage tips were directly placed into autosampler vials to elute the analyte with 60 µL of 50% MeCN, 0.1% FA (vol/vol) using a syringe. Eluates in the autosampler vial were dried in a SpeedVac, reconstituted in 3 µL 1% FA (vol/vol), and set up on the autosampler for injection on the mass spectrometer.

#### nLC-MS/MS

The samples in FA (1%) were loaded (2.5 µL) onto a 75 µm i.d. × 50 cm Acclaim PepMap 100 C18 RSLC column (Thermo Fisher Scientific) on an EASY *nano*LC (Thermo Fisher Scientific) at a constant pressure of 700 bar at 100% A (1% FA). Prior to sample loading, the column was equilibrated to 100% A for a total of 11 µL at 700 bar pressure. Peptide chromatography was initiated with mobile phase A (1% FA) containing 5% B (100% MeCN, 1% FA) for 1 min, then increased to 20% B over 19 min, to 32% B over 10 min, to 95% B over 1 min and held at 95% B for 19 min, with a flow rate of 250 nL/min. The data was acquired in data-independent acquisition (DIA) mode. The full-scan mass spectra were acquired with the Orbitrap mass analyzer with a scan range of m/z=150–1500 and a mass resolving power set to 70,000. DIA high-energy collisional dissociations were performed according to the inclusion list with a mass resolving power set to 17,500, an isolation width of 2 Da, and a normalized collision energy setting of 27. The maximum injection time was 60 ms for parent-ion analysis and product-ion analysis. The automatic gain control (AGC) was set at a target value of 3e6 ions for full MS scans and 2e5 ions for MS2. The inclusion list contained four peptide entries. The loop count was set at 30. Retention

time windows were set at 20 min. Each sample was run in two technical replicates, and the peak area ratio was averaged before concentration calculations of the peptides were conducted. Several quality control steps were conducted prior to running the in vivo samples. (1) Two technical replicates of a known concentration were injected and analyzed – an example table from four random experiments included in this manuscript is shown below. (2) The buffers used on the day of the experiment (aCSF and high K$^+$ buffer) were also tested for any contaminating Met-Enk or Leu-Enk signals by injecting two technical replicates for each buffer. Once these two criteria were met, the experiment was analyzed through the system. If step 1 failed, which happened a few times, the samples were frozen, and the machines were cleaned and restarted until the quality control measures were met. If step 2 failed, the samples were considered contaminated and not processed as part of the study. This quality control procedure was predetermined.

| Injected conc (aM) | QC before ME (aM) | QC after ME (aM) | Mean | STDEV | QC before LE (aM) | QC after LE (aM) | Mean | STDEV |
|---|---|---|---|---|---|---|---|---|
| 25 fM | 28 | 32 | 30.000 | 2.82842712 | 40 | 36 | 38 | 2.82842712 |
| 40 aM | 44 | 41 | 42.500 | 2.12132034 | 53 | 50 | 51.5 | 2.12132034 |
| 40 aM | 54 | 51 | 52.500 | 2.12132034 | 51 | 55 | 53 | 2.82842712 |
| 40 aM | 42 | 41 | 41.500 | 0.70710678 | 43 | 45 | 44 | 1.41421356 |

## Stable isotope dilution assay for measurement of enkephalins

Peptides were synthesized as high-purity natural abundance and stable isotope-labeled pairs for the development of a targeted proteomics method for enkephalins. Assays were developed as Tier 2 assays according to CPTAC guidelines (*Carr et al., 2014*). Co-eluting peak boundaries for the internal standard and endogenous peptides were manually refined using the Skyline software package (*MacLean et al., 2010*). Integrated peak areas for the three most intense fragment ions were then exported for determination of peak area ratios of endogenous to internal standard signal, which were used as a basis for determining abundance and quantifying changes in enkephalin levels in mouse studies.

### In vivo fiber photometry

#### Stereotaxic surgery

For the surgical procedure, *Penk*-Cre mice were briefly anesthetized in an induction chamber at 3% isoflurane before being placed in a stereotaxic frame (Kopf) secured with ear and bite bars. During the surgery, the isoflurane level was kept at 1.5–2%. Then, we performed a craniotomy and unilaterally injected 200–250 nL of AAV5.Syn.Flex.GCaMP6s.WPRE.SV40 using a Nanoject III (Drummond) (*Chen et al., 2013*) (Addgene: 100845-AAV5) at the following coordinates: +1.3 [AP], ±0.5 [ML], −5.0 mm [DV]. To allow for viral expression, the fiber photometry probes (5 mm shaft length, doric lenses) were implanted 3 weeks later at +1.3 [AP], ±0.5 [ML], −4.75 mm [DV]. Implanted probes were secured with dental adhesive (C&B Metabond kit) followed by cyanoacrylate (Loctite Super Glue Gel Control).

#### Fiber photometry recording procedure

Before recording, the implanted probes were tethered to an optic fiber using a ferrule sleeve (Doric, catalog no. ZR_2.5). Two light-emitting diodes (LEDs) were used to excite GCaMP6s. A 331 Hz sinusoidal LED light (Doric) of wavelength 470 nm 470 ± 20 nm was used for the Ca$^{2+}$-dependent signal. A 210 Hz sinusoidal LED light of wavelength 405 ± 10 nm (Doric, LED driver) was used to evoke the Ca$^{2+}$-independent signal and serve as the isosbestic control. GCaMP6s fluorescence traveled through the same optic fiber before being bandpass filtered (525 ± 25 nm, Doric, catalog no. FMC4), transduced by a femtowatt photoreceiver (New Focus Model 2151) and recorded by a real-time processor (TDT, RZ5P). The signals were extracted by the Synapse software (TDT). During the recording, two behavioral manipulations were introduced. The first was after a 10 min recording in a home cage environment. An experimenter handled the mouse while the recording was ongoing. The handling bout which mimicked traditional scruffing lasted about 3–5 s. The mouse was then let go, and the handling was repeated another two times in a single session with a minimum of 1–2 min between

handling bouts. Mice were habituated to this manipulation by being attached to the fiber photometry rig for 3–5 consecutive days prior to the experimental recording. Additionally, the same maneuver was employed when attaching/detaching the fiber photometry cord, so the mice were subjected to the same process several times. The second behavioral manipulation was introduced to the same mice a week after the first recording session. After a 10 min recording in the home cage environment, a weigh boat containing a kim wipe that had absorbed 3 mL of fox urine was placed in the cage as the recording proceeded. The weigh boat was removed after a minute of exploration, then reintroduced into the cage for another two times, separated by 5 min from the last exposure, during the session. In a subset of the animals (five of nine), the fox urine exposure was preceded by water exposure following the same process described above in the same recording session.

## Fiber photometry data analysis

Data analysis was performed through the Python toolbox GuPPy (*Sherathiya et al., 2021*). Briefly, the isosbestic control signal was used to correct for motion and photobleaching artifacts. The control channel is fitted to the signal channel using a least squares polynomial fit of degree 1. Transient detection occurred after using a 15 s moving window for thresholding transients. Then, high-amplitude events were identified, and their timestamp of occurrence during the session was reported. To determine the effect of handling or fox urine exposure on enkephalinergic cell activity, the event timestamp was used to observe changes in activity before and after the behavioral manipulation. The average of all trials is then calculated for repeated exposures and for exposures in multiple animals. The fiber photometry data was represented as z-scores.

## Immunohistochemistry and verification of probe placement

After the completion of the experiment, mice were anesthetized with a mixture of ketamine, xylazine, and acepromazine and transcardially perfused with ice-cold 4% paraformaldehyde in phosphate buffer. After the perfusion, brains were dissected and fixed for 24 hr in 4% paraformaldehyde at 4°C and then placed in 30% sucrose solution for cryoprotection for a minimum of 48 hr. The brains were then sliced into 30 µm sections, washed three times in PBS, and blocked in PBS containing 0.5% Triton X-100 and 5% goat serum for 1 hr. Following the blocking step, the slices were stained for the glial fibrillary acidic protein (GFAP), an astrocyte marker to determine glial scarring around the probe site (guinea pig anti-GFAP, synaptic systems) at 1:500 dilution and incubated at 4°C for 16 hr. The next day, the sections were washed three times with PBS and stained with the secondary antibody, mouse anti-guinea pig, Alexa Fluor 594 (Invitrogen) at 1:500 dilution and incubated for 2 hr at room temperature. After the incubation, the slices were washed three times in PBS, followed by two washes with PB. Finally, the slices were mounted on glass slides with HardSet Vectashield with DAPI staining (Vector Labs) and imaged using an epifluorescence microscope (Leica DM6 B). Correct probe placement in the NAcSh is represented in *Figure 3—figure supplement 1B*. For the verification of the injection and probe placement for fiber photometry, the same procedure described above was performed for tissue preparation. After slicing, the slices were washed in a 0.1% Triton for 1 hr in order to permeabilize the cells, followed by two 10 min PBS washes. The slices were then stained with NeuroTrace (Nissl) red (Invitrogen) to mark neurons by adding 5 µL to each well for 2 hr at room temperature, followed by three 10 min PBS washes and three 10 min PB washes. Finally, the slices were mounted using HardSet Vectashield (Vector Labs). GCaMP6s expression was determined by observing green fluorescent protein expression colocalized with NeuroTrace staining. Correct probe placement in the NAcSh is represented in *Figure 3—figure supplement 1D*. For misses in probe or viral placement, the data was not included in the manuscript as we only intended to obtain measurements from the NAcSh. This exclusion criterion was predetermined.

## Statistical analysis

We performed simple linear regression analysis on the forward and reverse curves and reported the line equations. For the forward curves, the regression was applied to the measured concentration of the light standard as the theoretical concentration was increased. For plotting purposes, we show the measured peak area ratios for the light standards in the forward curves. For the reverse curves, the regression was applied to the measured concentration of the heavy standard, as the theoretical concentration was varied. The proteomics data is represented as violin plots with individual points

depicted and dashed lines indicating quartiles. The middle-dashed lines represent the median. The violin plots were created using GraphPad Prism. The data was $\log_{10}$ transformed to reduce the skewness of the dataset caused by the variable range of concentrations measured across experiments/animals. Prior to log transformation, the measurements failed normality testing for a Gaussian distribution. After the log transformation, the data passed normality testing, which provided the rationale for the use of statistical analyses that assume normality. Two-way ANOVA testing with peptide (Met-Enk or Leu-Enk) and treatment (buffer or stress, for example) as the two independent variables. Post hoc testing was done using Šídák's multiple comparisons test, and the p-values for each of these analyses are shown in the figures (*Figures 1F and 2A*). A paired t-test was performed on the predator odor proteomic data before and after odor exposure to test the hypothesis that Met-Enk increases following exposure to predator odor (*Figure 3F*). These analyses were conducted using GraphPad Prism. To determine the linear relationship between the levels of Met-Enk to Leu-Enk in the same samples, we conducted simple linear regression analysis and reported the slopes of the lines. This is based on prior literature positing a relatively fixed ratio of Met-Enk to Leu-Enk, and we corroborate that here (*Comb et al., 1982*; *Hughes et al., 1997*; *Henderson et al., 1978*). For the fiber photometry data, the z-scores were calculated as described in using GuPPy, which is an open-source Python toolbox for fiber photometry analysis (*Sherathiya et al., 2021*). The z-score equation used in GuPPy is $z = (\Delta F/F - (\text{mean of } \Delta F/F)/\text{standard deviation of } \Delta F/F)$, where F refers to fluorescence of the GCaMP6s signal. For the averaged plots depicted in *Figures 2 and 3*, z-scores from all individual animals were averaged and shown as one trace with the SEM as the highlighted area. The purpose of the averaged traces is to show the extent of concordance of the response to experimenter handling and predator odor stress among animals, with the SEM demonstrating that variability. The heatmaps depict the individual responses of each animal. The heatmaps were plotted using Seaborn in Python, and mean traces were plotted using Matplotlib in Python.

## Acknowledgements

We would like to thank the Moron-Concepcion lab for providing us with the Enk-Cre mouse line. We would also like to thank Dr. Jordan McCall for his advice on data analysis and visualization as well as proofreading the manuscript. We would also like to thank the entirety of the proteomics core team, including Dr. Yiling Mi for her assistance. Thank you to the entirety of the Al-Hasani and McCall labs for their feedback on experimental design and analysis. Finally, thank you to Dr. Tim Holy, Dr. Robyn Klein, and Dr. Alexxai Kravitz for their feedback.

This work was supported by the National Institute on Drug Abuse R00 DA038725 (RA) and R21 DA048650 (RA). The NARSAD Young Investigator Grant from the Brain and Behavior Research Foundation, grant no. 28243 (RA). The WU-PSR is supported in part by the WU Institute of Clinical and Translational Sciences (NCATS UL1 TR000448), the Mass Spectrometry Research Resource (NIGMS P41 GM103422; R24GM136766), Siteman Comprehensive Cancer Center Support Grant (NCI P30 CA091842), and the Cognitive, Computational, and Systems Neuroscience Fellowship (MOM).

## Additional information

### Funding

| Funder | Grant reference number | Author |
| --- | --- | --- |
| National Institute on Drug Abuse | R00DA038725 | Ream Al-Hasani |
| National Institute on Drug Abuse | R21DA048650 | Ream Al-Hasani |
| National Institute of Neurological Disorders and Stroke | R01 NS135401 | Ream Al-Hasani |
| Brain and Behavior Research Foundation | 28243 | Ream Al-Hasani |

| Funder | Grant reference number | Author |
|---|---|---|
| National Center for Advancing Translational Sciences | UL1 TR000448 | Reid R Townsend |
| National Cancer Institute | P30 CA091842 | Reid R Townsend |
| Washington University in St Louis | Cognitive, Computational, and Systems Neuroscience Fellowship | Marwa O Mikati |
| Mass Spectrometry Research Resource | NIGMS P41 GM103422 | Reid R Townsend |
| Mass Spectrometry Research Resource | R24GM136766 | Reid R Townsend |

The funders had no role in study design, data collection and interpretation, or the decision to submit the work for publication.

## Author contributions

Marwa O Mikati, Data curation, Formal analysis, Investigation, Visualization, Methodology, Writing – original draft, Project administration, Writing – review and editing; Petra Erdmann-Gilmore, Software, Formal analysis, Validation, Methodology, Writing – original draft, Writing – review and editing; Rose Connors, Formal analysis, Methodology; Sineadh M Conway, Data curation, Formal analysis; Jim Malone, Robert W Sprung, Software, Formal analysis; Justin Woods, Data curation; Reid R Townsend, Conceptualization, Investigation, Writing – original draft, Project administration, Writing – review and editing; Ream Al-Hasani, Conceptualization, Resources, Data curation, Formal analysis, Supervision, Funding acquisition, Validation, Investigation, Visualization, Methodology, Writing – original draft, Project administration, Writing – review and editing

## Author ORCIDs

Marwa O Mikati  https://orcid.org/0000-0002-0914-8592
Jim Malone  https://orcid.org/0000-0003-2199-0456
Ream Al-Hasani  https://orcid.org/0000-0002-8781-6234

## Ethics

All surgical and behavioral procedures were approved by the Washington University School of Medicine animal care guidelines in accordance with federal animal use regulations IACUC protocol #23-0261. Mice were group housed and kept at a 12-hour light/dark cycle with ad libitum access to water and food. The holding facility was both temperature and humidity controlled.

Reviewer #1 (Public review): https://doi.org/10.7554/eLife.91609.3.sa1
Reviewer #2 (Public review): https://doi.org/10.7554/eLife.91609.3.sa2
Reviewer #3 (Public review): https://doi.org/10.7554/eLife.91609.3.sa3
Author response https://doi.org/10.7554/eLife.91609.3.sa4

# Additional files

## Supplementary files
MDAR checklist

## Data availability
The source data for all the figures in this study are included in the manuscript.

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
