## [Editor Report · eLife Assessment]

The authors adapt a previously-established method that permits detection of in vivo extracellular levels of two distinct enkephalin opioid peptides in response to stressful experiences in mice. The present study highlights the potential of measuring actual peptides by microdialysis-LC-MS. They use this approach in conjunction with fiber photometric calcium imaging to correlate enkephalin neuron activity and enkephalin release in response to repeated stress, providing **convincing** evidence that this improved approach can provide new insights into opioid signaling in-vivo. This **important** study provides a means to understand various behavioral states controlled by endogenous opioids and the nucleus accumbens, including hedonic and stress responses, in health and disease. This work will be of broad interest to the neuroscientific community.

---

## [Referee Report · Reviewer #1 (Public review)]

The present study by Mikati et al describes an improved method for in-vivo detection of enkephalin release and examines the impact of stress on activation of enkephalin neurons and enkephalin release in the nucleus accumbens (NAc). The authors refine their pipeline to measure met and leu enkephalin using liquid chromatography and mass spectrometry. The authors subsequently measure met and leu enkephalin in the NAc during stress induced by handling and fox urine, in addition to calcium activity of enkephalinergic cells using fiber photometry. The authors conclude that this improved tool for measuring enkephalin reveals experimenter handling stress-induced enkephalin release in the NAc that habituates and is dissociable from calcium activity of these cells, whose activity doesn't habituate. The authors subsequently show that NAc enkephalin neuron calcium activity does habituate to fox urine exposure, is activated by a novel weigh boat, and that fox urine acutely causes increases in met-enk levels, in some animals, as assessed by microdialysis. This study highlights a new approach to monitor two distinct enkephalins and more robust analytical approach for more sensitive detection of neuropeptides. The authors also provide a pipeline that potentially could aid in detection of other neuropeptides and increase our understanding of endogenous opioid neuropeptidergic control in health and disease.

---

## [Referee Report · Reviewer #2 (Public review)]

Summary:

The authors aimed to improve the detection of enkephalins, opioid peptides involved in pain modulation, reward, and stress. They used optogenetics, microdialysis and mass spectrometry to measure enkephalin release during acute stress in freely moving rodents. Their study provided detection and quantitation of enkephalins due to implementation of previously reported derivatization reaction combined with improved sample collection and offered insights into the dynamics and relationship between Met- and Leu-Enkephalin in the Nucleus Accumbens shell during stress.

Strengths:

A strength of this work is the quantitative Enk measurements resulted from an optimized microdialysis technique coupled with established derivatization approach and sensitive and quantitative nLC-MS measurements. This approach allowed basal and stimulated peptide release with higher temporal resolution, lower detection thresholds, and native-state endogenous peptide measurement.

Weaknesses:

The optimization of the previously published customizable microdialysis probe and the Met-Enk oxidation derivatization are included in the results, but these adjustments seem more like refinements or practical adaptations rather than significant innovations.

Another consideration is the use of log transformation for quantitation of peptides. Transforming data to achieve normality for parametric tests does not guarantee that all assumptions of normality are met, especially in small and variable datasets reported here. Visual checks like QQ-plots can help verify the appropriateness of transformations. In biological contexts, log transformation can obscure the relationship between measured values and underlying processes.

---

## [Referee Report · Reviewer #3 (Public review)]

Summary:

This important paper describes improvements to the measurement of enkephalins in vivo using microdialysis and LC-MS. The key improvement is oxidation of met- to prevent having a mix of reduced and oxidized methionine the sample which make quantification more difficult. It then shows measurements of enkephalins in the nucleus accumbens in two different stress situations-handling and exposure to predator odor. It also reports the ratio of released met- and leu-enkephalin matching that expected from digestion of proenkephalin. Measurements are also made by photometry of Ca2+ changes for the fox odor stressor. Some key takeaways are: (1) reliable measurement of met-enkephalin, significance of directly measuring peptides as opposed to proxy measurements, and the opening of a new avenue into research of enkephalins due to stress based on these direct measurements.

Strengths:

- Improved methods for measurement of enkephalins in vivo

- Compelling examples of using this method

- Opening a new area of looking at stress responses through the lens of enkephalin concentrations

Comments on revisions:

This revision has been improved upon in most ways. As I mentioned in the original review, there is a great deal of work here on showing the capability of measuring met- and leu-enk in different contexts. There is a technical improvement in the control of met oxidation which likely improves the detection of met-enk.

---

## [Author Response]

The following is the authors’ response to the original reviews.

**Public Reviews:**

**Reviewer #1 (Public Review):**

Thank you for your constructive feedback and recognition of our work. We followed your suggestion and improved the accuracy of the language used to interpret some of our findings.

Summary:The present study by Mikati et al demonstrates an improved method for in-vivo detection of enkephalin release and studies the impact of stress on the activation of enkephalin neurons and enkephalin release in the nucleus accumbens (NAc). The authors refine their pipeline to measure met and leu enkephalin using liquid chromatography and mass spectrometry. The authors subsequently measured met and leu enkephalin in the NAc during stress induced by handling, and fox urine, in addition to calcium activity of enkephalinergic cells using fiber photometry. The authors conclude that this improved tool for measuring enkephalin reveals experimenter handling stress-induced enkephalin release in the NAc that habituates and is dissociable from the calcium activity of these cells, whose activity doesn't habituate. The authors subsequently show that NAc enkephalin neuron calcium activity does habituate to fox urine exposure, is activated by a novel weigh boat, and that fox urine acutely causes increases in met-enk levels, in some animals, as assessed by microdialysis.Strengths:A new approach to monitoring two distinct enkephalins and a more robust analytical approach for more sensitive detection of neuropeptides. A pipeline that potentially could help for the detection of other neuropeptides.Weaknesses:Some of the interpretations are not fully supported by the existing data or would require further testing to draw those conclusions. This can be addressed by appropriately tampering down interpretations and acknowledging other limitations the authors did not cover brought by procedural differences between experiments.

We have taken time to go through the manuscript ensuring we are more detailed and precise with our interpretations as well as appropriately acknowledging limitations.

**Reviewer #2 (Public Review):**

Thank you for your constructive and thorough assessment of our work. In our revised manuscript, we adjusted the text to reflect the references you mentioned regarding the methionine oxidation procedure. Additionally, we expanded the methods section to include the key details of the statistical tests and procedures that you outlined.

Summary:The authors aimed to improve the detection of enkephalins, opioid peptides involved in pain modulation, reward, and stress. They used optogenetics, microdialysis, and mass spectrometry to measure enkephalin release during acute stress in freely moving rodents. Their study provided better detection of enkephalins due to the implementation of previously reported derivatization reaction combined with improved sample collection and offered insights into the dynamics and relationship between Met- and Leu-Enkephalin in the Nucleus Accumbens shell during stress.Strengths:A strength of this work is the enhanced opioid peptide detection resulting from an improved microdialysis technique coupled with an established derivatization approach and sensitive and quantitative nLC-MS measurements. These improvements allowed basal and stimulated peptide release with higher temporal resolution, lower detection thresholds, and native-state endogenous peptide measurement.Weaknesses:The draft incorrectly credits itself for the development of an oxidation method for the stabilization of Met- and Leu-Enk peptides. The use of hydrogen peroxide reaction for the oxidation of Met-Enk in various biological samples, including brain regions, has been reported previously, although the protocols may slightly vary. Specifically, the manuscript writes about "a critical discovery in the stabilization of enkephalin detection" and that they have "developed a method of methionine stabilization." Those statements are incorrect and the preceding papers that relied on hydrogen peroxide reaction for oxidation of Met-Enk and HPLC for quantification of oxidized Enk forms should be cited. One suggested example is Finn A, Agren G, Bjellerup P, Vedin I, Lundeberg T. Production and characterization of antibodies for the specific determination of the opioid peptide Met5-Enkephalin-Arg6-Phe7. Scand J Clin Lab Invest. 2004;64(1):49-56. doi: 10.1080/00365510410004119. PMID: 15025428.

Thank you for highlighting this. It was not our intention to imply that we developed the oxidation method, rather that we were able improve the detection of metenkephalin by oxidation of the methionine without compromising the detection resolution of leu-enkephalin, enabling the simultaneous detection of both peptides. We have addressed this is the manuscript and included the suggested citation.

Another suggestion for this draft is to make the method section more comprehensive by adding information on specific tools and parameters used for statistical analysis:

(1) Need to define "proteomics data" and explain whether calculations were performed on EIC for each m/z corresponding to specific peptides or as a batch processing for all detected peptides, from which only select findings are reported here. What type of data normalization was used, and other relevant details of data handling? Explain how Met- and Leu-Enk were identified from DIA data, and what tools were used.

Thank you for pointing out this source of confusion. We believe it is because we use a different DIA method than is typically used in other literature. Briefly, we use a DIA method with the targeted inclusion list to ensure MS2 triggering as opposed to using large isolation widths to capture all precursors for fragmentation, as is typically done with MS1 features. For our method, MS2 is triggered based on the 4 selected m/z values (heavy and light versions of Leu and Met-Enkephalin peptides) at specific retention time windows with isolation width of 2 Da; regardless of the intensity of MS1 of the peptides.

(2) Simple Linear Regression Analysis: The text mentions that simple linear regression analysis was performed on forward and reverse curves, and line equations were reported, but it lacks details such as the specific variables being regressed (although figures have labels) and any associated statistical parameters (e.g., R-squared values).

Additional detail about the linear regression process was added to the methods section, please see lines 614-618. The R squared values are also now shown on the figure.

‘For the forward curves, the regression was applied to the measured concentration of the light standard as the theoretical concentration was increased. For plotting purposes, we show the measured peak area ratios for the light standards in the forward curves. For the reverse curves, the regression was applied to the measured concentration of the heavy standard, as the theoretical concentration was varied.’

(3) Violin Plots: The proteomics data is represented as violin plots with quartiles and median lines. This visual representation is mentioned, but there is no detail regarding the software/tools used for creating these plots.

We used Graphpad Prism to create these plots. This detail has been added to the statistical analysis section. See line 630.

(4) Log Transformation: The text states that the data was log-transformed to reduce skewness, which is a common data preprocessing step. However, it does not specify the base of the logarithm used or any information about the distribution before and after transformation.

We have added the requested details about the log transformation, and how the data looked before and after, into the statistical analysis section. We followed convention that the use of log is generally base 10 unless otherwise specified as natural log (base 2) or a different base. See lines 622-625

‘The data was log10 transformed to reduce the skewness of the dataset caused by the variable range of concentrations measured across experiments/animals. Prior to log transformation, the measurements failed normality testing for a Gaussian distribution. After the log transformation, the data passed normality testing, which provided the rationale for the use of statistical analyses that assume normality.’

(5) Two-Way ANOVA: Two-way ANOVA was conducted with peptide and treatment as independent variables. This analysis is described, but there is no information regarding the software or statistical tests used, p-values, post-hoc tests, or any results of this analysis.

Information about the two-way ANOVA analysis has been added to the statistical analysis section. Additionally, more detailed information has been added to the figure legends about the statistical results. Please see lines 625-628.

‘Two-way ANOVA testing with peptide (Met-Enk or Leu-Enk) and treatment (buffer or stress for example) as the two independent variables. Post-hoc testing was done using Šídák's multiple comparisons test and the p values for each of these analyses are shown in the figures (Figs. 1F, 2A).’

(6) Paired T-Test: A paired t-test was performed on predator odor proteomic data before and after treatment. This step is mentioned, but specific details like sample sizes, and the hypothesis being tested are not provided.

The sample size is included in the figure legend to which we have included a reference. We have also included the following text to highlight the purpose of this test. See lines 628-630

A paired t-test was performed on the predator odor proteomic data before and after odor exposure to test that hypothesis that Met-Enk increases following exposure to predator odor (Fig. 3F). These analyses were conducted using Graphpad Prism.

(7) Correlation Analysis: The text mentions a simple linear regression analysis to correlate the levels of Met-Enk and Leu-Enk and reports the slopes. However, details such as correlation coefficients, and p-values are missing.

We apologize for the use of the word correlation as we think it may have caused some confusion and have adjusted the language accordingly. Since this was a linear regression analysis, there is no correlation coefficient. The slope of the fitted line is reported on the figures to show the fitted values of Met-Enk to Leu-Enk.

(8) Fiber Photometry Data: Z-scores were calculated for fiber photometry data, and a reference to a cited source is provided. This section lacks details about the calculation of zscores, and their use in the analysis.

These details have been added to the statistical analysis section. See lines 634-637

‘For the fiber photometry data, the z-scores were calculated as described in using GuPPy which is an open-source python toolbox for fiber photometry analysis. The z-score equation used in GuPPy is z=(DF/F-(mean of DF/F)/standard deviation of DF/F) where F refers to fluorescence of the GCaMP6s signal.’

(9) Averaged Plots: Z-scores from individual animals were averaged and represented with SEM. It is briefly described, but more details about the number of animals, the purpose of averaging, and the significance of SEM are needed.

We have added additional information about the averaging process in the statistical analysis section. See lines 639-643.

‘The purpose of the averaged traces is to show the extent of concordance of the response to experimenter handling and predator odor stress among animals with the SEM demonstrating that variability. The heatmaps depict the individual responses of each animal. The heatmaps were plotted using Seaborn in Python and mean traces were plotted using Matplotlib in Python.’

A more comprehensive and objective interpretation of results could enhance the overall quality of the paper.

We have taken this opportunity to improve our manuscript following comments from all the reviewers that we hope has resulted in a manuscript with a more objective interpretation of results.

**Reviewer #3 (Public Review):**

Thank you for your thoughtful review of our work. To clarify some of the points you raised, we revised the manuscript to include more detail on how we distinguish between the oxidized endogenous and standard signal, as well as refine the language concerning the spatial resolution. We also edited the manuscript regarding the concentration measurements. We conducted technical replicates, so we appreciate you raising this point and clarify that in the main text.

Summary:This important paper describes improvements to the measurement of enkephalins in vivo using microdialysis and LC-MS. The key improvement is the oxidation of met- to prevent having a mix of reduced and oxidized methionine in the sample which makes quantification more difficult. It then shows measurements of enkephalins in the nucleus accumbens in two different stress situations - handling and exposure to predator odor. It also reports the ratio of released met- and leu-enkephalin matching what is expected from the digestion of proenkephalin. Measurements are also made by photometry of Ca2+ changes for the fox odor stressor. Some key takeaways are the reliable measurement of met-enkephalin, the significance of directly measuring peptides as opposed to proxy measurements, and the opening of a new avenue into the research of enkephalins due to stress based on these direct measurements.Strengths:-Improved methods for measurement of enkephalins in vivo.-Compelling examples of using this method.-Opening a new area of looking at stress responses through the lens of enkephalin concentrations.Weaknesses:(1) It is not clear if oxidized met-enk is endogenous or not and this method eliminates being able to discern that.

We clarified our wording in the text copied below to provide an explanation on how we distinguish between the two. Even after oxidation, the standard signal has a higher m/z ratio due to the presence of the Carbon and Nitrogen isotopes as described in the Chemicals section of the methods ‘For Met Enkephalin, a fully labeled L-Phenylalanine (^13^C_9_, ^15^N) was added (YGGFM). The resulting mass shift between the endogenous (light) and heavy isotope-labeled peptide are 7Da and 10Da, respectively.’, so they can still be differentiated from the endogenous signal. We have clarified the language in the results section. See lines 82-87.

‘After each sample collection, we add a consistent known concentration of isotopically labeled internal standard of Met-Enk and Leu-Enk of 40 amol/sample to the collected ISF for the accurate identification and quantification of endogenous peptide. These internal standards have a different mass/charge (m/z) ratio than endogenous Met- and Leu-Enk. Thus, we can identify true endogenous signal for Met-Enk and Leu-Enk (Suppl Fig. 1A,C) versus noise, interfering signals, and standard signal (Suppl. Fig. 1B,D).’

(2) It is not clear if the spatial resolution is really better as claimed since other probes of similar dimensions have been used.

Apologies for any confusion here. To clarify we primarily state that our approach improves temporal resolution and in a few cases refer to improved spatiotemporal resolution, which we believe we show. The dimensions of the microdialysis probe used in these experiments allow us to target the nucleus accumbens shell and as well as being smaller – especially at the membrane level - than a fiber photometry probe.

(3) Claims of having the first concentration measurement are not quite accurate.

Thank you for your feedback. To clarify, we do not claim that we have the first concentration measurements, rather we are the first to quantify the ratio of Met-Enk to Leu-Enk in vivo in freely behaving animals in the NAcSh.

(4) Without a report of technical replicates, the reliability of the method is not as wellevaluated as might be expected.

We have added these details in the methods section, please see lines 521-530.

‘Each sample was run in two technical replicates and the peak area ratio was averaged before concentration calculations of the peptides were conducted. Several quality control steps were conducted prior to running the in vivo samples. (1) Two technical replicates of a known concentration were injected and analyzed – an example table from 4 random experiments included in this manuscript is shown below. (2) The buffers used on the day of the experiment (aCSF and high K+ buffer) were also tested for any contaminating Met-Enk or Leu-Enk signals by injecting two technical replicates for each buffer. Once these two criteria were met, the experiment was analyzed through the system. If either step failed, which happened a few times, the samples were frozen and the machines were cleaned and restarted until the quality control measures were met.’

**Recommendations For The Authors:**

**Reviewer #1 (Recommendations For The Authors):**
• The authors should provide appropriate citations of a study that has validated the Enkephalin-Cre mouse line in the nucleus accumbens or provide verification experiments if they have any available.

Thank you for your comment. We have added a reference validating the Enk-Cre mouse line in the nucleus accumbens to the methods section and is copied here.

D.C. Castro, C.S. Oswell, E.T. Zhang, C.E. Pedersen, S.C. Piantadosi, M.A. Rossi, A.C. Hunker, A. Guglin, J.A. Morón, L.S. Zweifel, G.D. Stuber, M.R. Bruchas, An endogenous opioid circuit determines state-dependent reward consumption, Nature 2021 598:7882 598 (2021) 646–651. https://doi.org/10.1038/s41586-02104013-0.

• Better definition of the labels y1,y2,b3 in Figures 1 and S1 would be useful. I may have missed it but it wasn't described in methods, results, or legends.

Thank you for this comment. We have added this information to Fig.1 legend ‘Y1, y2, b3 refer to the different elution fragments resulting from Met-Enk during LC-MS.

• It is interesting that the ratio of KCl-evoked release is what changes differentially for Met- vs Leu. Leu enk increases to the range of met-enk. There is non-detectable or approaching being non-detectable leu-enk (below the 40 amol / sample limit of quantification) in most of the subjects that become apparent and approach basal levels of met-enkephalin. This suggests that the K+ evoked response may be more pronounced for leu-enk. This is something that should be considered for further analysis and should be discussed.

Thank you for this astute observation, and you make a great point. We have added some discussion of this finding in the results and discussion sections see lines 111112 and lines 253-257.

‘Interestingly, Leu-Enk showed a greater fold change compared to baseline than did Met-Enk with the fold changes being 28 and 7 respectively based on the data in Fig.1F.’

‘We also noted that Leu-Enk showed a greater fold increase relative to baseline after depolarization with high K+ buffer as compared to Met-Enk. This may be due to increased Leu-Enk packaging in dense core vesicles compared to Met-Enk or due to the fact that there are two distinct precursor sources for Leu-Enk, namely both proenkephalin and prodynorphin while Met-Enk is mostly cleaved from proenkephalin (see Table 1 [48]).’

• For example in 2E, it would be helpful to label in the graph axis what samples correspond to the manipulation and also in the text provide the reader with the sample numbers. The authors interpret the relationship between the last two samples of baseline and posthandling stress as the following in the figure legend "the concentration released in later samples is affected; such influence suggests that there is regulation of the maximum amount of peptide to be released in NAcSh. E. The negative correlation in panel d is reversed by using a high K+ buffer to evoke Met-Enk release, suggesting that the limited release observed in D is due to modulation of peptide release rather than depletion of reserves." However, the correlations are similar between 2D and E and it appears that two mice are mediating the difference between the two groups. The appropriate statistical analysis would be to compare the regressions of the two groups. Statistics for the high K+ (and all other graphs where appropriate) need to be reported, including the r2 and p-value.

Thank you for your constructive critique. To elucidate the effect of high K+, we have plotted the regression line and reported the slope for Fig. 2E. Notably, the slope is reduced by a factor of 2 and appears to be driven by a large subset of the animals. The statistics for the high K+ graph are shown on the figure (Fig 1F) which test the hypothesis of whether high K+ leads to the release of Leu-Enk and Met-Enk respectively compared to baseline with aCSF. We have added the test statistics to the figure legend for additional clarity. Fig. 1G has no statistics because it is only there to elucidate the ratio between Met-Enk and Leu-Enk in the same samples. We did not test any hypotheses related to whether there are differences between their levels as that is not relevant to our question. The correlation on the same data is depicted in Fig. 1H, and we have added the R^2^ value per your request.

• The interpretation that handling stress induces enkephalin release from microdialysis experiments is also confounded by other factors. For instance, from the methods, it appears that mice were connected and sample collection started 30 min after surgery, therefore recovery from anesthesia is also a confounding variable, among other technical aspects, such as equilibration of the interstitial fluid to the aCSF running through the probe that is acting as a transmitter and extracellular molecule "sink". Did the authors try to handle the mice post hookup similar to what was done with photometry to have a more direct comparison to photometry experiments? This procedural difference, recording from recently surgerized animals (microdialysis) vs well-recovered animals with photometry should be mentioned in addition to the other caveats the authors mention.

Thank you for your comment. We are aware of this technical limitation, and it is largely why we sought to conduct the fiber photometry experiments to get at the same question. As you requested, we have included additional language in the discussion to acknowledge this limitation and how we chose to address it by measuring calcium activity in the enkephalinergic neurons, which would presumably be the same cell population whose release we are quantifying using microdialysis. See lines 262-273.

‘Our findings showed a robust increase in peptide release at the beginning of experiments, which we interpreted as due to experimenter handling stress that directly precedes microdialysis collections. However, there are other technical limitations to consider such as the fact that we were collecting samples from mice that were recently operated on. Another consideration is that the circulation of aCSF through the probe may cause a sudden shift in oncotic and hydrostatic forces, leading to increased peptide release to the extracellular space. As such, we wanted to examine our findings using a different technique, so we chose to record calcium activity from enkephalinergic neurons - the same cell population leading to peptide release. Using fiber photometry, we showed that enkephalinergic neurons are activated by stress exposure, both experimenter handling and fox odor, thereby adding more evidence to suggest that enkephalinergic neurons are activated by stress exposure which could explain the heightened peptide levels at the beginning of microdialysis experiments.’

• The authors should provide more details on handling stress manipulation during photometry. For photometry what was the duration of the handling bout, what was the interval between handling events, and can the authors provide a description of what handling entailed? Were mice habituated to handling days before doing photometry recording experiments?

Thank you for your suggestion. We have addressed all of your points in the methods section. See lines 564-570.

‘The handling bout which mimicked traditional scruffing lasted about 3-5 seconds. The mouse was then let go and the handling was repeated another two times in a single session with a minimum of 1-2 minutes between handling bouts. Mice were habituated to this manipulation by being attached to the fiber photometry rig, for 3-5 consecutive days prior to the experimental recording. Additionally, the same maneuver was employed when attaching/detaching the fiber photometry cord, so the mice were subjected to the same process several times.’

• For the novel weigh boat experiments, the authors should explicitly state when these experiments were done in relation to the fox urine, was it a different session or the same session? Were they the same animals? Statements like the following (line 251) imply it was done in the same animals in the same session but it should be clarified in the methods "We also showed using fiber photometry that the novelty of the introduction of a foreign object to the cage, before adding fox odor, was sufficient to activate enkephalinergic neurons."

As shown in supplementary figure 4, individual animal data is shown for both water and fox urine exposure (overlaid) to depict whether there were differences in their responses to each manipulation – in the same animal. And yes, you are correct, the animals were first exposed to water 3 times in the recording session and then exposed to fox urine 3 times in the same session. We have added that to the methods section describing in vivo fiber photometry. See lines 575-576.

• Statistical testing would be needed to affirm the conclusions the authors draw from the fox urine and novel weigh boat experiments. For example, it shows stats that the response attenuates, that it is not different between fox urine and novel (it looks like the response is stronger to the fox urine when looking at the individual animals), etc. These data look clear but stats are formally needed. Formal statistics are also missing in other parts of the manuscript where conclusions are drawn from the data but direct statistical comparisons are not included (e.g. Fig 2.G-I).

The photometry data is shown as z-scores which is a formal statistical analysis. ANOVA would be inappropriate to run to compare z-scores. We understand that this is erroneously done in fiber photometry literature, however, it remains incorrect. The z-scores alone provide all the information needed about the deviation from baseline. We understand that this is not immediately clear to readers, and we thank you for allowing us to explain why this is the case. We have added test statistics to figure legends where hypothesis testing was done and *p*-values were reported.

• Did the authors try to present the animals with repeated fox urine exposure to see if this habituates like the photometry?

No, we did not do that experiment due to the constrained timing within which we had to run our microdialysis/LC-MS timeline, but it is a great point for future exploration.

• It would be useful to present the time course of the odor experiment for the microdialysis experiment.

The timeline is shown in Fig.1a and Fig.3e. To reiterate, each sample is 13 minutes long.

• Can the authors determine if differences in behavior (e.g. excessive avoidance in animals with with one type of response) or microdialysis probe location dictate whether animals fall into categories of increased release, no release, or no-detection? From the breakdown, it looks like it is almost equally split into three parts but the authors' descriptions of this split are somewhat misleading (line 210). " The response to predator odor varies appreciably: although most animals show increased Met-Enk release after fox odor exposure, some show continued release with no elevation in Met-Enk levels, and a minority show no detectable release".

Thank you for your constructive feedback. We do not believe the difference in behavior is correlated with probe placement. The hit map can be found in suppl. Fig 3 and shows that all mice included in the manuscript had probes in the NAcSh. We purposely did not distinguish between dorsal and ventral because of our 1 mm membrane would make it hard to presume exclusive sampling from one subregion. That is a great point though, and we have thought about it extensively for future studies. We have edited the language to reflect the almost even split of responses for Met-Enk and appreciate you pointing that out.

• Overall, given the inconsistencies in experimental design and overall caveats associated, I think the authors are unable to draw reasonable conclusions from the repeated stressor experiments and something they should either consider is not trying to draw strong conclusions from these observations or perform additional experiments that provide the grounds to derive those conclusions.

We have included additional language on the caveats of our study, and our use of a dual approach using fiber photometry and microdialysis was largely driven by a

desire to offer additional support of our conclusions. We expected pushback about our conclusions, so we wanted to offer a secondary analysis using a different technique to test our hypothesis. To be honest the tone of this comment and content is not particularly constructive (especially for trainees) nor does it offer a space to realistically address anything. This work took multiple years to optimize, it was led by a graduate student, and required a multidisciplinary team. As highlighted, we believe it offers an important contribution to the literature and pushes the field of peptide detection forward.

**Reviewer #2 (Recommendations For The Authors):**
A more comprehensive and objective interpretation of results could enhance the overall quality of the paper. The manuscript contains statements like "we are the first to confirm," which can be challenging to substantiate and may not significantly enhance the paper. It's essential to ensure that novelty statements are well-founded. For example, the release of enkephalins from other brain regions after stress exposure is well-documented but not addressed in the paper. Similarly, the role of the NA shell in stress has been extensively studied but lacks coverage in this manuscript.

We have edited the language to reflect your feedback. We have also included relevant literature expanding on the demonstrated roles of enkephalins in the literature. We would like to note that most studies have focused on chronic stress, and we were particularly interested in acute stress. See lines 129-134.

‘These studies have included regions such as the locus coeruleus, the ventral medulla, the basolateral nucleus of the amygdala, and the nucleus accumbens core and shell. Studies using global knockout of enkephalins have shown varying responses to chronic stress interventions where male knockout mice showed resistance to chronic mild stress in one study, while another study showed that enkephalin-knockout mice showed delayed termination of corticosteroid release. [33,34]’Finally, not a weakness but a clarification suggestion: the method description mentions the use of 1% FA in the sample reconstitution solution and LC solvents, which is an unusually high concentration of acid. If this concentration is intentional for maintaining the peptides' oxidation state, it would be beneficial to mention this in the text to assist readers who might want to replicate the method.

This is correct and has been clarified in the methods section

**Reviewer #3 (Recommendations For The Authors):**
-The Abstract should state the critical improvements that are made. Also, quantify the improvements in spatiotemporal resolution.

Thank you for your comment. We have edited the abstract to reflect this.

- The use of "amol/sample" as concentration is less informative than an SI units (e.g., pM concentration) and should be changed. Especially since the volume used was the same for in vivo sampling experiments.

Thank you for your comment. We chose to report amol/sample because we are measuring such a small concentration and wanted to account for any slight errors in volume that can make drastic differences on reported concentrations especially since samples are dried and resuspended.

-Please check this sentence: "After each collection, the samples were spiked with 2 µL of 12.5 fM isotopically labeled Met-Enkephalin and Leu-Enkephalin" This dilution would yield a concentration of ~2 fM. In a 12 uL sample, that would be ~0.02 amol, well below the detection limit. (note that fM would femtomolar concentration and fmol would be femtomoles added).-"liquid chromatography/mass spectrometry (LC-MS) [9-12]"... Reference 9 is a RIA analysis paper, not LC-MS as stated.

Thank you for catching these. We have corrected the unit and citation.

-Given that improvements in temporal resolution are claimed, the lack of time course data with a time axis is surprising. Rather, data for baseline and during treatment appear to be combined in different plots. Time course plots of individuals and group averages would be informative.

Due to the expected variability between individual animal time course data, where for example, we measure detectable levels in one sample followed by no detection, it was very difficult to combine data across time. Therefore, to maximize data inclusion from all animals that showed baseline measurements and responses to individual manipulations, we opted to report snapshot data. Our improvement in temporal resolution refers to the duration of each sample rather than continuous sampling, so those two are unrelated. Thank you for your feedback and allowing us to clarify this.

- I do not understand this claim "We use custom-made microdialysis probes, intentionally modified so they are similar in size to commonly used fiber photometry probes to avoid extensive tissue damage caused by traditional microdialysis probes (Fig. 1B)." The probes used are 320 um OD and 1 mm long. This is not an uncommon size of microdialysis probes and indeed many are smaller, so is their probe really causing less damage than traditional probes?

Thank you for your comment. We are only trying to make the point that the tissue damage from these probes is comparable to commonly used fiber photometry probes. We only point that out because tissue damage is used as a point to dissuade the usage of microdialysis in some literature, and we just wanted to disambiguate that. We have clarified the statement you pointed out.

-The oxidation procedure is a good idea, as mentioned above. It would be interesting to compare met-enk with and without the oxidation procedure to see how much it affects the result (I would not say this is necessary though). It is not uncommon to add antioxidants to avoid losses like this. Also, it should be acknowledged that the treatment does prevent the detection of any in vivo oxidation, perhaps that is important in met-enk metabolism?

The comparison between oxidized and unoxidized Met-Enk detection is in figure 1C.

-It would be a best practice to report the standard deviation of signal for technical replicates (say near in vivo concentrations) of standards and repeated analysis of a dialysate sample to be able to understand the variability associated with this method. Similarly, an averaged basal concentration from all rats.

Thank you for your comment. We have included a table showing example quality control standard injections from 4 randomly selected experiments included in the manuscript that were run before and after each experiment and descriptive statistics associated with these technical replicates. We also added some detail to the methods section to describe how quality control is done. See lines 521-530.

‘Each sample was run in two technical replicates and the peak area ratio was averaged before concentration calculations of the peptides were conducted. Several quality control steps were conducted prior to running the in vivo samples. (1) Two technical replicates of a known concentration were injected and analyzed – an example table from 4 random experiments included in this manuscript is shown below. (2) The buffers used on the day of the experiment (aCSF and high K+ buffer) were also tested for any contaminating Met-Enk or Leu-Enk signals by injecting two technical replicates for each buffer. Once these two criteria were met, the experiment was analyzed through the system. If either step failed, which happened a few times, the samples were frozen and the machines were cleaned and restarted until the quality control measures were met.’

EDITORS NOTEShould you choose to revise your manuscript, please include full statistical reporting including exact p-values wherever possible alongside the summary statistics (test statistic and df) and 95% confidence intervals. These should be reported for all key questions and not only when the p-value is less than 0.05.

Thank you for your suggestion. We have included more detail about statistical analysis in the figure legends per this comment and reviewer comments.